# CRISPR-Cas in *Acinetobacter baumannii* Contributes to Antibiotic Susceptibility by Targeting Endogenous *AbaI*

Yuhang Wang,[a,b] Jie Yang,[a] Xiaoli Sun,[b] Mengying Li,[a,c] Pengyu Zhang,[a,b] Zhongtian Zhu,[a,d] Hongmei Jiao,[a,d] Tingting Guo,[a,b,c] Guocai Li[a,b,c,d]

[a]Department of Microbiology, Institute of Translational Medicine, Medical College, Yangzhou University, Yangzhou, PR China

[b]Jiangsu Key Laboratory of Experimental & Translational Non-coding RNA Research, Yangzhou, PR China

[c]Jiangsu Key Laboratory of Zoonosis/Jiangsu Co-Innovation Center for Prevention and Control of Important Animal Infectious Diseases and Zoonoses, Yangzhou University, Yangzhou, PR China

[d]Department of Laboratory Medicine, Affiliated Hospital of Yangzhou University, Yangzhou, PR China

**ABSTRACT** *Acinetobacter baumannii* is a well-known human opportunistic pathogen in nosocomial infections, and the emergence of multidrug-resistant *Acinetobacter baumannii* has become a complex problem for clinical anti-infective treatments. The ways this organism obtains multidrug resistance phenotype include horizontal gene transfer and other mechanisms, such as altered targets, decreased permeability, increased enzyme production, overexpression of efflux pumps, metabolic changes, and biofilm formation. A CRISPR-Cas system generally consists of a CRISPR array and one or more operons of *cas* genes, which can restrict horizontal gene transfer in bacteria. Nevertheless, it is unclear how CRISPR-Cas systems regulate antibiotic resistance in *Acinetobacter baumannii*. Thus, we sought to assess how CRISPR-Cas affects biofilm formation, membrane permeability, efflux pump, reactive oxygen species, and quorum sensing to clarify further the mechanism of CRISPR-Cas regulation of *Acinetobacter baumannii* antibiotic resistance. In the clinical isolate AB43, which has a complete I-Fb CRISPR-Cas system, we discovered that the Cas3 nuclease of this type I-F CRISPR-Cas system regulates *Acinetobacter baumannii* quorum sensing and has a unique function in changing drug resistance. As a result of quorum sensing, synthase *abaI* is reduced, allowing efflux pumps to decrease, biofilm formation to become weaker, reactive oxygen species to generate, and drug resistance to decrease in response to CRISPR-Cas activity. These observations suggest that the CRISPR-Cas system targeting endogenous *abaI* may boost bacterial antibiotic sensitivity.

**IMPORTANCE** CRISPR-Cas systems are vital for genome editing, bacterial virulence, and antibiotic resistance. How CRISPR-Cas systems regulate antibiotic resistance in *Acinetobacter baumannii* is almost wholly unknown. In this study, we reveal that the quorum sensing regulator *abaI* mRNA was a primary target of the I-Fb CRISPR-Cas system and the cleavage activity of Cas3 was the most critical factor in regulating *abaI* mRNA degradation. These results advance our understanding of how CRISPR-Cas systems inhibit drug resistance. However, the mechanism of endogenous targeting of *abaI* by CRISPR-Cas needs to be further explored.

**KEYWORDS** *Acinetobacter baumannii*, CRISPR-Cas, *abaI*, antibiotic susceptibility

*A*cinetobacter baumannii (*A. baumannii*) is a lactose-nonfermenting, Gram-negative bacterium that can survive in natural environments and hospitals because it has little need for living conditions (1). In 2017, the World Health Organization (WHO) listed carbapenem-resistant *A. baumannii* as a significant priority, implying that new antibiotics are urgently needed to combat this species (2). Overuse of antibiotics has caused the emergence of multidrug resistance in *A. baumannii*, as a consequence of either horizontal gene transfer (3) or other factors, such as altered targets, decreased membrane

Address correspondence to Guocai Li, gcli@yzu.edu.cn.

The authors declare no conflict of interest.

permeability (4), increased production of degrading enzymes (5, 6), overexpression of efflux pumps (7), metabolic changes (8), biofilm formation (9, 10), or increased nutrient sequestration mechanisms (11).

The CRISPR-Cas system is an acquired intrinsic immune defense system of bacteria that maintains the stability of the bacterial genome by resisting the invasion of exogenous genetic material such as plasmids, phages, etc. (12). A total of 39% of sequenced bacterial and 88% of archaeal genomes contain CRISPR-Cas systems (13), with approximately 70% of pathogenic bacteria containing type I CRISPR-Cas systems (14). Based on the composition of Cas proteins, CRISPR-Cas systems are currently classified into two main categories, six types (I–VI), and more than 30 subtypes (15, 16). Type I-F is the most prevalent CRISPR-Cas system in *A. baumannii* (17). The I-Fb CRISPR-Cas system consists of Cas1, Cas3, Csy1, Csy2, Csy3, Csy4, and CRISPR arrays (18, 19). Studies have revealed that Cas3 cleaves invading RNA (20) by acting as a single-strand DNA nuclease and an ATP-dependent helicase (21). This cascade is composed of Csy1 to Csy4 proteins, wherein Csy4 processes crRNA transcripts, and the Csy1 to Csy3 proteins are necessary to stabilize the crRNA generated by Csy4 (22).

In addition to being an essential part of the prokaryotic immune system that prevents viral infection, the CRISPR-Cas systems also have various roles in physiology, such as boosting bacterial virulence and countering antibiotic resistance (23–25). For example, recent studies have shown that *A. baumannii* uses the I-Fb CRISPR-Cas system to stop the spread of antibiotic resistance genes (26). Our recent research also demonstrated that the I-Fb CRISPR-Cas-related gene *Csy1* in AB43 was upregulated when treated with most antibiotics and only downregulated when treated with doxycycline and kanamycin as an antibiotic pressure (27). However, it is unclear why this phenomenon occurs and its specific mechanism. CRISPR-Cas also incorporates cues for cell population density into its regulation.

Using quorum sensing (QS), bacteria can communicate with each other through extracellular signals, allowing them to obtain information about their surroundings, densities, and metabolic activities (28). In *A. baumannii*, AbaR (QS receptor) and AbaI (QS synthase), through a self-secreted signaling molecule called N-Acyl homoserine lactone (AHL), control the expression of specific phenotypes such as motility, antibiotic resistance, survival, and biofilm formation (29, 30). More specifically, QS activates type I-F CRISPR-Cas expression and CRISPR adaptation in *P. aeruginosa*, allowing CRISPR-Cas activity to increase in tandem with bacterial cell density (31). When bacterial populations have a high cell density and a high chance of phage infection and dissemination, this method guarantees maximal CRISPR-Cas activity to exert its immune defense. On the other hand, when bacteria invade mammalian host cells, the Cas3 of *P. aeruginosa* recognizes and cleaves the QS regulator *lasR* mRNA to enhance its virulence (32). The details of how CRISPR-Cas systems work to regulate antibiotic resistance remain unknown.

We, therefore, investigated the role of the Type I-Fb CRISPR-Cas system in *A. baumannii* in modulating QS operation as it affects bacterial resistance to antibiotics. We found that the Type I-Fb CRISPR-Cas of *A. baumannii* degraded mRNA of the QS master regulator *abaI*, and the cleavage activity of Cas3 was the most critical factor in regulating *abaI* mRNA degradation. Our study shows how CRISPR-Cas systems mechanistically regulate antibiotic resistance in *A. baumannii*.

## RESULTS

**Detection of CRISPR-Cas systems in *A. baumannii* isolates.** Of the 245 randomly collected *A. baumannii* clinical isolates, no isolate was susceptible to all 24 antibiotics (Fig. 1A). Among the 245 *A. baumannii* isolates tested, 16/245 (6.53%) and 20/245 (8.16%) isolates were resistant to only one or two of the nine classes of antibiotics tested, respectively. Specifically, 209/245 (85.31%) were classified as multidrug-resistant (MDR; resistance to three or more classes of antibiotics). In *A. baumannii*, most identified CRISPR-Cas systems were type I-F, and the Cas operon was composed of six

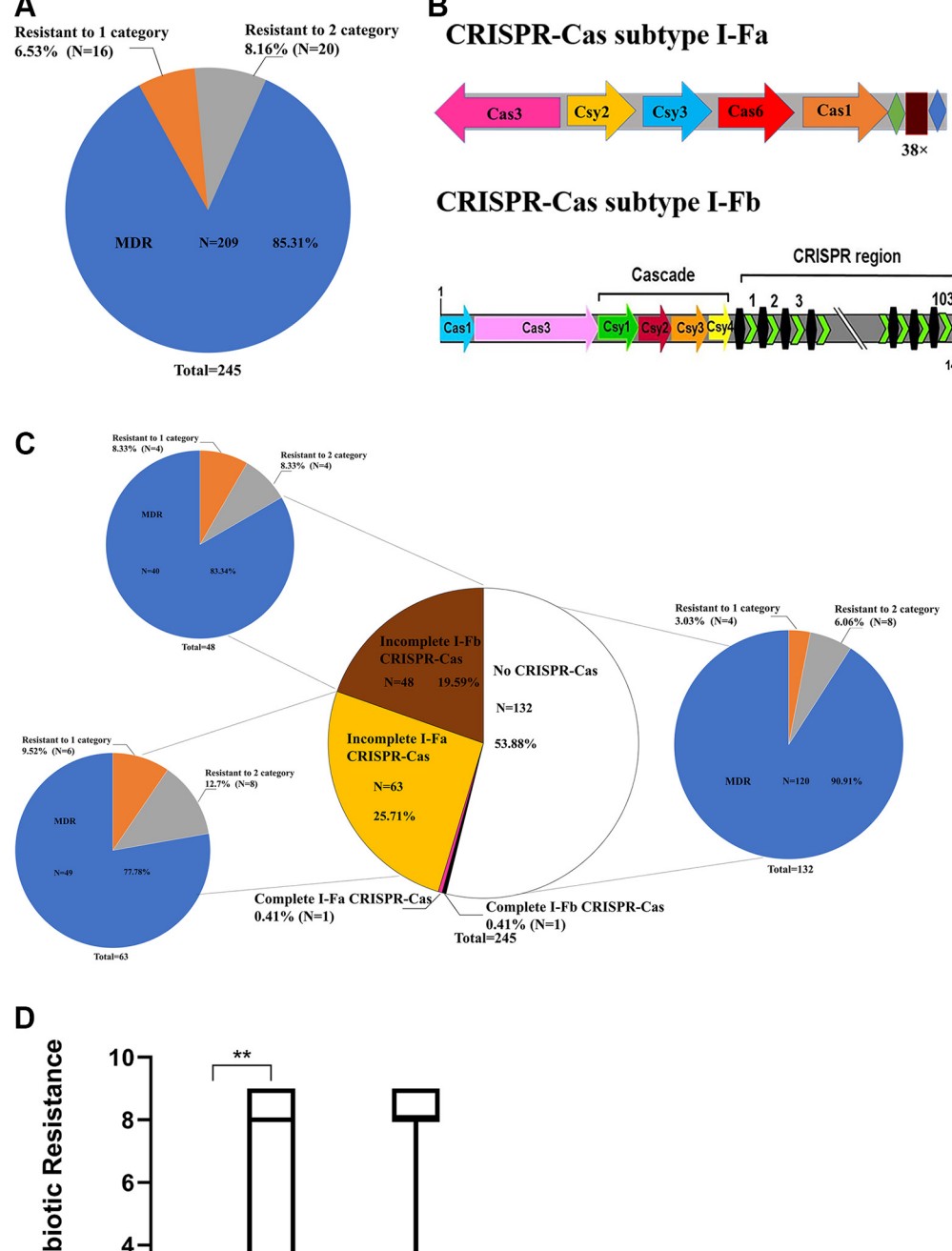

**FIG 1** Characterization of *A. baumannii* clinical isolates. (A) Of a total 245 *A. baumannii* clinical strains, 16 (6.53%) were resistant to one category of antibiotic (orange), 20 (8.16%) were resistant to two categories of antibiotics (gray), and 209 (85.31%) were resistant to three or more categories of antibiotics (blue). (B) Schematic of I-Fa and I-Fb CRISPR-Cas system in *A. baumannii*. (C) Of the total 245 all *A. baumannii* clinical strains, 1 (0.41%) had a complete I-Fa CRISPR-Cas system (red), 1 (0.41%) had a complete I-Fb CRISPR-Cas system (black), 63 (25.71%) had an

**TABLE 1** Relationship between the *cas* genes and the drug resistance phenotype in 245 clinical *A. baumannii* strains[a]

| | Positive | | Negative | |
|---|---|---|---|---|
| Cas | No. of MDR isolates | % (n = 245) | No. of MDR isolates | % (n = 245) |
| I-Fa-*cas1* | 49 | 20% | 160 | 65.31% |
| I-Fa-*cas3* | 7 | 2.86% | 202 | 82.45% |
| I-Fa-*csy2* | 22 | 8.98% | 187 | 76.33% |
| I-Fa-*csy3* | 1 | 0.41% | 208 | 84.9% |
| I-Fa-*cas6* | 28 | 11.43% | 181 | 73.88% |
| I-Fb-*cas1* | 40 | 16.33% | 169 | 68.98% |
| I-Fb-*cas3* | 5 | 2.04% | 204 | 83.27% |
| I-Fb-*csy1* | 14 | 5.71% | 195 | 79.59% |
| I-Fb-*csy2* | 9 | 3.67% | 200 | 81.63% |
| I-Fb-*csy3* | 8 | 3.27% | 201 | 82.04% |
| I-Fb-*csy4* | 34 | 13.88% | 175 | 71.43% |

[a]MDR, multidrug resistance (resistance to three or more classes of antibiotics).

to seven genes (17) (Fig. 1B). We screened for the CRISPR-Cas systems using PCR in the collected 245 *A. baumannii* clinical isolates. Of the 64 isolates with the I-Fa CRISPR-Cas system, only one isolate (resistant to one category of antibiotics tested) had a complete I-Fa CRISPR-Cas system, while the other 63 (49 MDR) I-Fa CRISPR-Cas-positive clinical isolates were incomplete. Similarly, of 49 isolates with the I-Fb CRISPR-Cas system, only one isolate (resistant to one category of antibiotics tested) was found to have a complete I-Fb CRISPR-Cas system. The other 48 (40 MDR) I-Fb CRISPR-Cas positive clinical isolates were incomplete. MLST of the 113 CRISPR-Cas-positive isolates revealed 40 different sequence types (STs). The most prevalent ST was ST1145 (25/113, 22.12%), followed by ST195 (15/113, 13.27%), ST1696 (12/113, 10.62%), and ST1417 (9/113, 7.96%) (Table S3). Furthermore, only these four STs have both I-Fa and I-Fb CRISPR-Cas systems, and no crossover distribution has been found in other STs. Additionally, 132 isolates were not positive for a CRISPR-Cas system, 4/132 (3.03%) and 8/132 (6.06%) isolates were resistant to only one or two of the nine classes of antibiotics tested, respectively, and 120/132 (90.91%) were classified as MDR (Fig. 1C).

The PCR results showed that among these 245 strains, most strains with incomplete or without CRISPR-Cas systems were MDR (Fig. 1D). To explore whether drug resistance in *A. baumannii* possessing an incomplete CRISPR-Cas system is associated with a specific Cas protein, we statistically analyzed the relationship between drug resistance phenotypes and *cas* genes in these 113 CRISPR-Cas-positive strains, and the results are shown in Table 1. We found that all *cas* gene-negative strains had significantly higher resistance rates than positive strains. I-Fa *csy3*-negative or I-Fb *cas3*-negative had the highest resistance rates in I-Fa and I-Fb *cas* gene-negative strains, respectively. In this regard, it is speculated that the incomplete CRISPR-Cas system, especially the loss of I-Fa *csy3* and I-Fb *cas3*, may affect antibiotic resistance in *A. baumannii*. The I-Fb CRISPR-Cas system is highly conserved and can prevent the horizontal transfer of junction elements (33); we identified a complete I-Fb CRISPR-Cas system in AB43. This strain resisted one category of antibiotics tested (Table 2). Thus, we wanted to explore the critical *cas* gene that inhibits drug resistance and the possible mechanisms in the I-Fb CRISPR-Cas system.

**FIG 1** Legend (Continued)
incomplete I-Fa CRISPR-Cas system (yellow), 48 (19.59%) had an incomplete I-Fb CRISPR-Cas system (brown), and 132 (53.88%) were CRISPR-Cas negative (white). Of a total 63 incomplete I-Fa CRISPR-Cas *A. baumannii* clinical strains, 49 (77.78%) were resistant to three or more classes of antibiotic (blue), while 6 (9.52%) were resistant to one category of antibiotic (orange), and 8 (12.7%) were resistant to two category antibiotics (gray). Of a total 48 incomplete I-Fb CRISPR-Cas *A. baumannii* clinical strains, 40 (83.34%) were resistant to three or more classes of antibiotics (blue), 4 (8.33%) were resistant to one category of antibiotic (orange), and 4 (8.33%) were resistant to two categories of antibiotics (gray). Of 132 *A. baumannii* clinical strains without CRISPR-Cas, 120 (90.91%) were resistant to three or more classes of antibiotics (blue), 4 (3.03%) were resistant to one category of antibiotic (orange), and 8 (6.06%) were resistant to two categories of antibiotics (gray). (D) Correlation between the numbers of antibiotics each *A. baumannii* isolate was resistant to. **, *P*-value was significant (*P*-value < 0.01), calculated by the U Mann-Whitney test.

**TABLE 2** Drug resistance of AB43-derived strains[a]

| Antimicrobial | Drug | AB43 | AB43Δ*crispr-cas* | AB43Δ*abal* | AB43Δ*crispr-cas-abal* |
|---|---|---|---|---|---|
| Penicillin | Piperacillin | R | R | **I** | **I** |
| β-lactam/β-lactamase inhibitor combinations | Ampicillin-sulbactam | S | **R** | S | S |
| | Piperacillin-tazobactam | S | **R** | S | S |
| | Ticarcillin-clavulanate | S | **R** | S | S |
| Cephems | Cefotaxime | I | **R** | **S** | **S** |
| | Ceftazidime | S | **R** | S | S |
| | Ceftriaxone | I | **R** | **S** | **S** |
| | Cefepime | S | **R** | S | S |
| Carbapenems | Imipenem | S | **R** | S | S |
| | Meropenem | S | **R** | S | S |
| Aminoglycosides | Amikacin | S | **R** | S | S |
| | Gentamicin | S | **R** | S | S |
| | Tobramycin | S | **R** | S | S |
| Fluoroquinolones | Ciprofloxacin | S | **R** | S | S |
| | Levofloxacin | S | **R** | S | S |
| | Gatifloxacin | S | **I** | S | S |
| Lipopeptides | Polymyxin B | S | S | S | S |
| Tetracyclines | Tetracycline | S | **R** | S | S |
| | Doxycycline | S | **R** | S | S |
| | Minocycline | S | S | S | S |
| Folate pathway inhibitors | Sulfisoxazole | S | **R** | S | S |
| | Roxithromycin | S | **R** | S | S |
| | Rifampicin | S | **R** | S | S |
| | Chloramphenicol | S | **R** | S | S |

[a]R, resistance; I, intermediate; S, sensitivity. Boldface indicates antimicrobial susceptibility results change.

**Complete CRISPR-Cas system represses antibiotic resistance in *A. baumannii*.**
We first constructed one mutant by deleting the CRISPR-Cas cluster (AB43Δ*crispr-cas*)
and restored the locus in a deletion mutant background (AB43Δ*crispr-cas*/p*crispr-cas*).
Compared to the wild type (WT), the deletion of the *crispr-cas* and its restoration did
not affect *A. baumannii* growth (Fig. S1). We further constructed a series of single-locus
mutants derived from AB43, including *cas1*, *cas3*, *csy1*, *csy2*, *csy3*, *csy4*, and *crispr*, as
well as their corresponding complemented strains. Antimicrobial susceptibility tests for
the WT, deletion mutants, and complemented strains were evaluated using 24 drugs
belonging to nine major antibiotic types. While the AB43 and all the gene rescue
mutants were susceptible to most antibiotics, deletion of any component or total abol-
ishment of the CRISPR-Cas system rendered AB43 significantly resistant to most tested
drugs (Table 2 and Table S5–S11).

**Transcriptomic analysis of the entire CRISPR-Cas knockout mutant.** After dem-
onstrating that the entire CRISPR-Cas system was required to suppress antibiotic resistance
in *A. baumannii*, we sought to clarify the molecular mechanisms of this phenomenon. To
address this issue, we performed transcriptomic analyses of AB43 and AB43Δ*crispr-cas*. As
shown in Fig. 2A, and a total of 1,403 genes showed a significant difference in expression
between these two strains, with 189 genes being upregulated and 1,214 genes being
downregulated in the AB43Δ*crispr-cas* strain. KEGG enrichment analysis revealed that
these genes were related to the two-component system (TCS) and various bacterial me-
tabolism-related pathways (Fig. 2C and D). Specifically, these genes with repressed expres-
sion were correlated with the TCS, while those with increased expression were involved in
multidrug efflux pumps (Fig. 2B). Particularly, genes associated with the ATP-binding cas-
sette (ABC) transporters and multidrug efflux pumps increased dramatically, implying
more of this essential function was present in the AB43Δ*crispr-cas* strain.

**CRISPR-Cas inhibits *A. baumannii* efflux pumps.** To further validate the CRISPR-
Cas regulation of efflux pumps, we next monitored the efflux activity in different AB43-
derived genotypes using ethidium bromide (EtBr) as a fluorescent probe. The results
showed that increased efflux of EtBr occurred in deletion mutants (Fig. 3A, Fig. S2A).
Similarly, qRT-PCR results showed that the mRNA expression of genes related to the
efflux pump, such as ABC transporters (*macB* and *emrB*) (34), the major facilitator

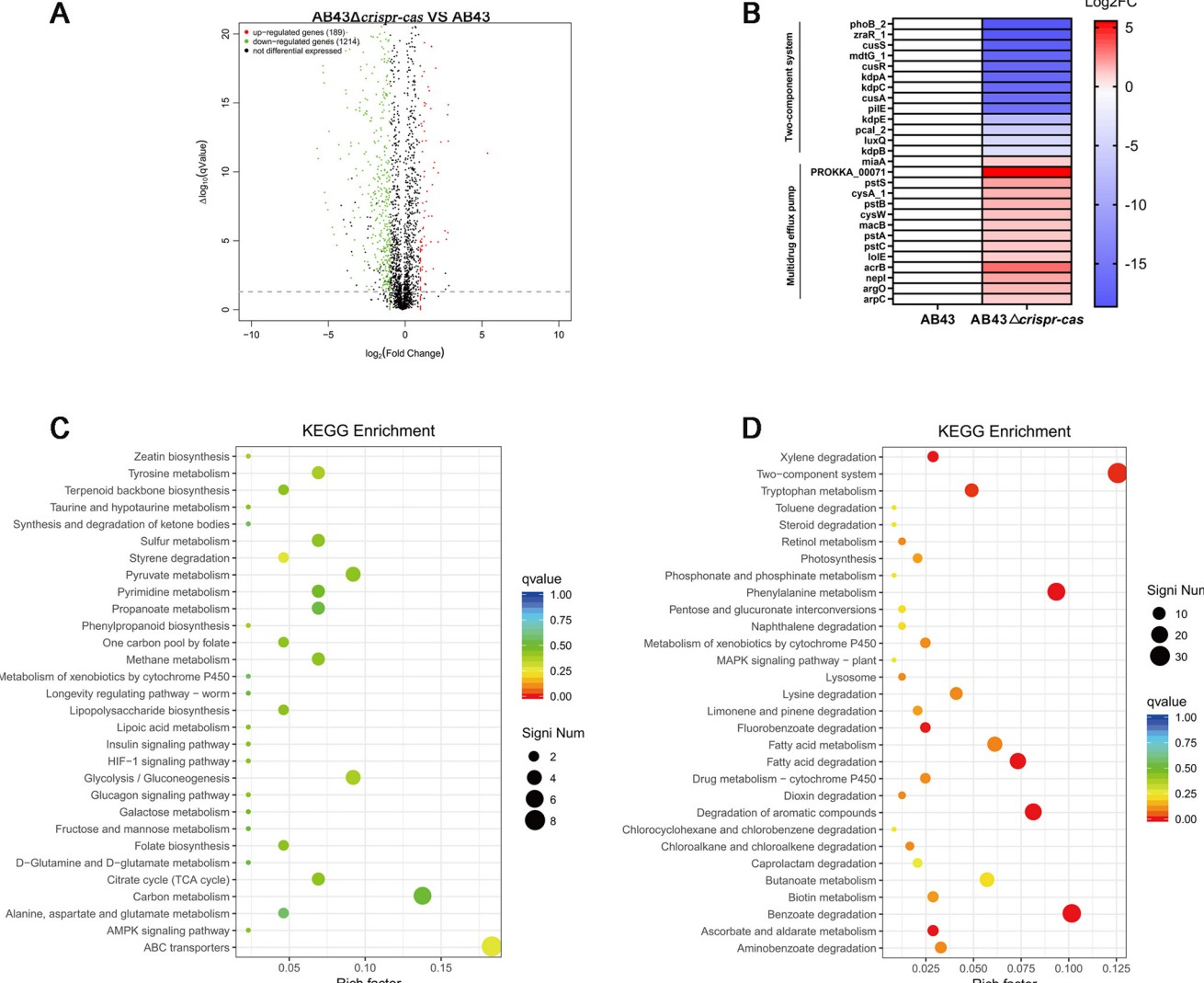

**FIG 2** Transcriptomic analysis of an entire *crispr-cas* knockout mutant. (A) Volcano plot annotations of differentially expressed genes (DEGs) in a AB43Δ*crispr-cas* strain. The *x-* and *y* axes in indicate changes in expression and statistically significant degrees, respectively. The adjusted *P*-value (*P* < 0.05; Student's *t* test with Benjamini–Hochberg false discovery rate adjustment) and |log2Fold change|≥1 were applied for determining DEG significance. (B) Selected differential expression genes involved in two-component systems and multidrug efflux pumps. An analysis of KEGG enrichment gene expression of (C) upregulated DEGs; and (D) downregulated DEGs. Data from three biological replicates were used.

superfamily (MFS) (*craA*, *rpoB*, *tetB*, and *abaQ*), the resistance-nodulation-cell division (RND) superfamily (*adeB* and *adeG*), and the small multidrug resistance (SMR) protein family (*abeS* and *abeM*) in the knockout strain were elevated (Fig. 3B and C) (35). The efflux pump superfamilies use energy from the proton motive force (PMF), except for the ABC superfamilies, which utilize energy from ATP hydrolysis to facilitate the efflux of substances within a cell (36). We used the pH-sensitive fluorescent probe BCECF-AM to evaluate the PMF of these strains. AB43 deletion mutants increased fluorescence compared to a WT strain (Fig. 3D, Fig. S2B). As the PMF drives ATP synthesis (37), the intracellular levels of ATP also significantly increased in deletion strains (Fig. 3E, Fig. S2C). Altogether, these results showed that the entire CRISPR-Cas system significantly represses the mRNA expression of ABC transporters and efflux pump genes and inhibits the energy required for efflux.

**CRISPR-Cas deletion impacts *A. baumannii* biofilm formation and membrane permeability.** Next, we explored other resistance mechanisms besides efflux pump activity. As biofilm formation is a well-known mechanism for antibiotic resistance in *A.*

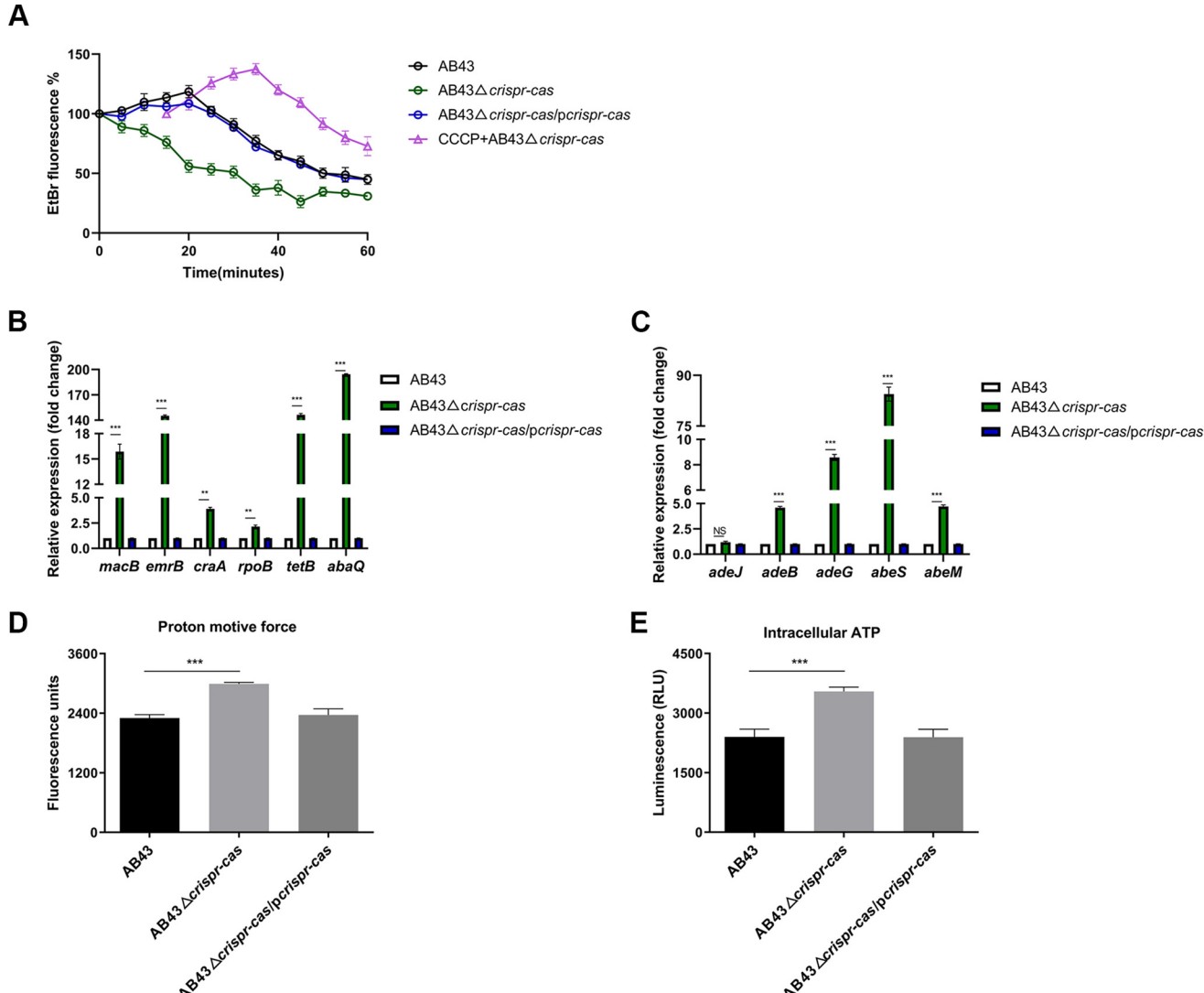

**FIG 3** CRISPR-Cas inhibits *A. baumannii* efflux pumps. (A) Bacteria were coincubated with EtBr (8 µg/mL final concentration) or the known efflux pump inhibitor CCCP ($10^{-4}$ M) at 37°C. Then, EtBr efflux from cells was monitored with an excitation wavelength of 530 nm and an emission wavelength of 600 nm for 60 min. (B and C) RNA was isolated when AB43, AB43Δ*crispr-cas*, and the complementary strain AB43Δ*crispr-cas*/p*crispr-cas* grew to an $OD_{600}$ of 1.0 in liquid LB medium. Transcripts of indicated efflux pump-related genes (B) ABC transporters (*macB*, *emrB*) and MFS (*craA*, *rpoB*, *tetB*, and *abaQ*); (C) RND (*adeB* and *adeG*) and SMR (*abeS* and *abeM*) in AB43Δ*crispr-cas* strains were then quantified by qRT-PCR. (D) CRISPR-Cas decreased PMF based on the fluorescence intensity of BCECF-AM-probed *A. baumannii* cells. (E) A luciferin-luciferase bioluminescence assay has decreased the production of intracellular ATP in AB43. Significance was evaluated using nonparametric one-way ANOVA (**, $P < 0.01$; ***, $P < 0.001$; NS, not significant). All data are presented as the mean ± standard error of the mean (SEM).

*baumannii* (38), we used crystal violet to detect the deletion strains' biofilm formation. Our results showed that AB43Δ*crispr-cas* formed a significantly more robust biofilm than WT or its complemented strains (Fig. 4A). Similarly, AB43Δ*cas1*, AB43Δ*cas3*, AB43Δ*csy1*, AB43Δ*csy2*, AB43Δ*csy3*, AB43Δ*csy4*, and AB43Δ*crispr* all developed higher levels of bacterial biofilms than their respective gene complementary strains (Fig. S3A).

Since multidrug resistance in *A. baumannii* often requires an inner membrane permease (39), we hypothesized that the CRISPR-Cas system might inhibit the membrane permeability of AB43. To test this, we used the fluorescent probe propidium iodide (PI) to test membrane permeability (40), and we found that the deletion mutants tested exhibited significantly lower membrane permeabilities than the WT strain and complementary strains (Fig. 4B, Fig. S3B). We also assessed the effects of CRISPR-Cas on the permeability of the outer membrane (OM) via fluorescence intensity analysis. N-Phenyl-1-naphthylamine (NPN), a hydrophobic fluorescent probe that releases fluorescence when interacting with

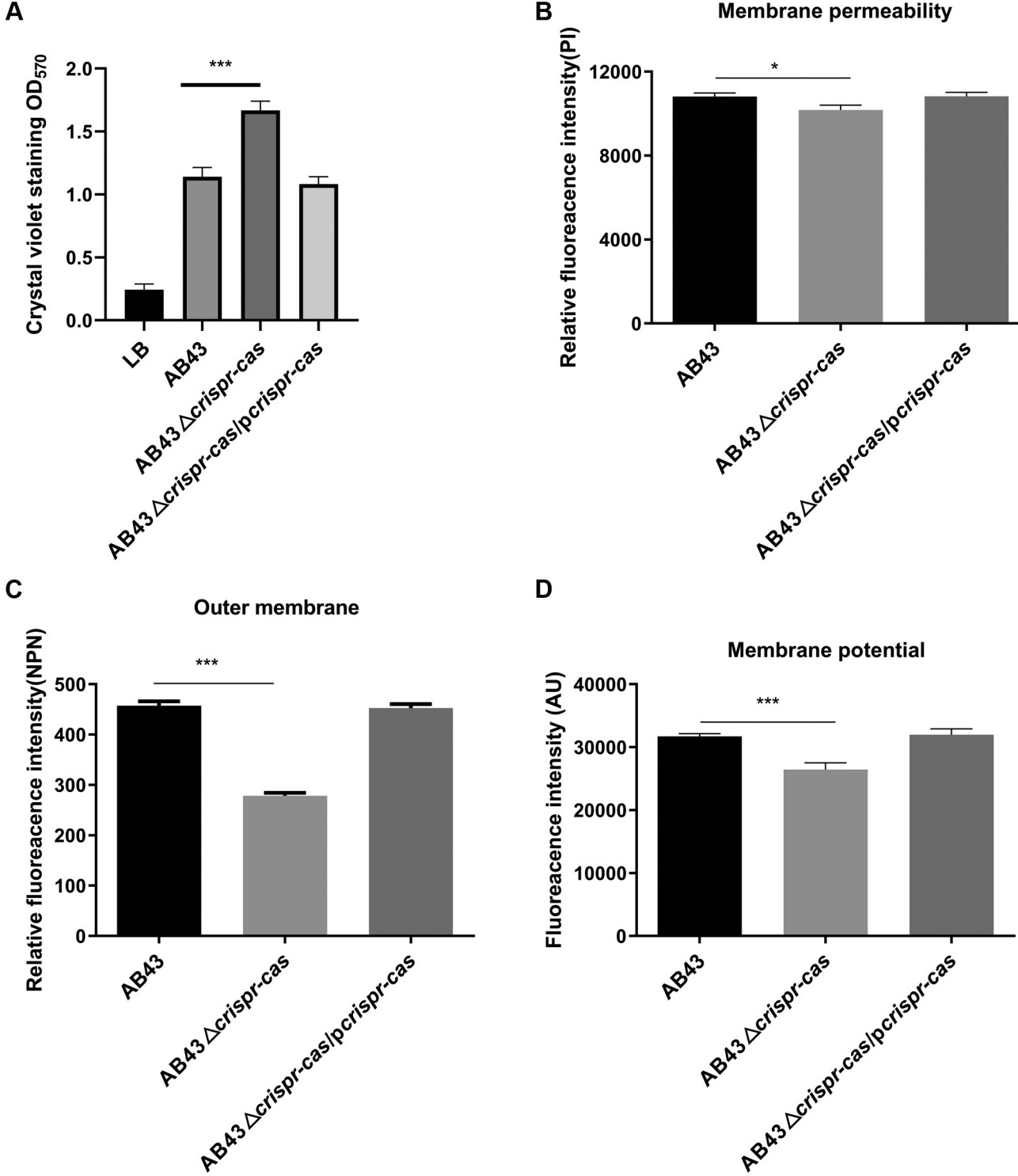

**FIG 4** *Crispr-cas* deletion impact on *A. baumannii* biofilm formation and membrane permeability. (A) Measurement of AB43Δ*crispr-cas* biofilm biomass by crystal violet staining. *A. baumannii* strains were cultured in sterile 96-well microtiter plates, and the absorbance was recorded at 570 nm. (B) *Crispr-cas* deletion reduced the membrane permeability of *A. baumannii*. (C) *Crispr-cas* deletion reduced the outer membrane permeability of *A. baumannii*. (D) *Crispr-cas* deletion dissipates membrane potential. DiSC3(5) dye was applied to determine the membrane potential. The dissipated membrane potential of *A. baumannii* was measured with an excitation wavelength of 622 nm and an emission wavelength of 670 nm. Mean values of three independent experiments and SEM values are shown. *, $P < 0.05$; ***, $P < 0.001$.

the hydrophobic parts of a phospholipid bilayer (41), was used to assess the permeability of the OM. In line with previous membrane permeability results, the knockout of any CRISPR-Cas component decreased the OM permeability of AB43 (Fig. 4C, Fig. S3C). DiSC3 (5) was exploited to evaluate the bacterial membrane potential. We found this significantly reduced deletion mutants compared to AB43 and the complemented strain (Fig. 4D, Fig. S3D). These results indicated that dysfunction of the CRISPR-Cas system could enhance AB43 biofilm biomass and dampen bacterial membrane permeability, which shows a synergistic effect with efflux pumps.

**CRISPR-Cas contributes to reactive oxygen species generation in *A. baumannii*.** It was recently reported that the production of reactive oxygen species (ROS) is essential in the sensitivity of *A. baumannii* to antibiotics (42). To that end, we wanted to explore whether ROS was involved in the process of CRISPR-Cas inhibition of drug resistance. The results showed that the deletion strains had reduced generation of total ROS (Fig. 5A, Fig. S4A) and increased intracellular superoxide dismutase (SOD) activity compared with AB43 (Fig. 5B, Fig. S4B). In cells, ROS included hydrogen peroxide ($H_2O_2$), superoxide ($O2\cdot-$), and hydroxyl radicals ($OH\cdot$). It was also interesting to note that CRISPR-Cas significantly promoted the production of $H_2O_2$ (Fig. 5C, Fig. S4C). Moreover, inhibiting tricarboxylic acid (TCA) cycle activity reduced ROS formation, and this TCA cycle activity required NAD. To further determine whether CRISPR-Cas impacts the TCA cycle, the $NAD^+/NADH$ ratio in deletion strains was determined. Consistent with the results above, the deletion strains had significantly increased $NAD^+/NADH$ ratios, indicating a reduced TCA cycle in mutants (Fig. 5D, Fig. S4D). We concluded that the CRISPR-Cas system increases ROS damage by promoting the TCA cycle and reducing the activity of the bacterial antioxidant system.

**CRISPR-Cas represses drug resistance by targeting *abaI* mRNA.** Next, we sought to determine which transcripts or genes were being disabled by CRISPR-Cas upstream of these different phenotypic changes. Previous studies have suggested that the I-F CRISPR-Cas system in *P. aeruginosa* can target the bacterial quorum-sensing regulator *lasR* mRNA and reduce bacterial virulence (32). Moreover, the QS system regulates bacterial luminescence, toxin production, disinfectants tolerance, motility, biofilm formation, spore formation, and drug resistance (43). Intriguingly, we found that the QS synthase gene *abaI* contains one region matching the CRISPR array. The region from nucleotides (nt) 29 to 39 in *abaI* is partly matched with spacer 101 and repeats (Fig. 6A). Thus, we utilized a specific bacterial biosensor (44), the *A. tumafaciens* KYC55 strain, to determine the amount of AHL activity in AB43-derived strains. As shown in Fig. 6B, the AHL activity of AB43Δ*crispr-cas* was stronger than the WT strain. Compared with AB43, more than half (5/8, 62.5%) of the deletion strains: AB43Δ*crispr-cas*, AB43Δ*cas3*, AB43Δ*csy1*, AB43Δ*csy3*, AB43Δ*csy4*, showed significantly elevated of *abaI* mRNA (Fig. 6C). On the contrary, the *abaI* transcript level in each mutant complemented by the corresponding component were similar to those in AB43 (Fig. S5A). Then, we constructed *abaI* deletion mutant and tested its drug resistance. Compared to AB43, AB43Δ*abaI* was sensitive to all 23 tested drugs except piperacillin (Table 2). Moreover, qRT-PCR results showed that the mRNA expression of genes related to drug resistance, such as MFS (*craA* and *rpoB*), the RND superfamily (*adeJ*), TCS (*pmrA*, *pmrB*, *adeS*, *bfmS*, and *bfmR*) (45), biofilm formation (*ompA* and *lpsB*) (46), clinically significant cephalosporin resistance gene ($bla_{OXA-51-like}$) (47), and others (48–51) in the AB43Δ*abaI* were reduced (Fig. 7). Beyond this, the biological traits associated with drug resistance are also altered (Fig. S6). Furthermore, double knockdown of *crispr-cas* and *abaI* was comparable to the *abaI* knockdown alone in antibiotic resistance profiles (Table 2, Fig. 7, and Fig. S6).

We investigated the CRISPR-Cas system to understand which gene was necessary to regulate *abaI*. There were still eight *A. baumannii* with incomplete I-Fb CRISPR-Cas systems resistant to one or two category antibiotics that were tested in our collection of clinical isolates. We determined the *cas* gene expression of these eight strains and found that it compared with AB43; the *cas* gene expression level of these eight strains was significantly increased (Fig. S7). The initial results indicated that *cas3*, *csy1*, *csy3*, and *csy4* were primarily involved in regulating *abaI* mRNA expression, and the AB43Δ*cas3* expression of *abaI* was uppermost among the single *cas* gene knockdown

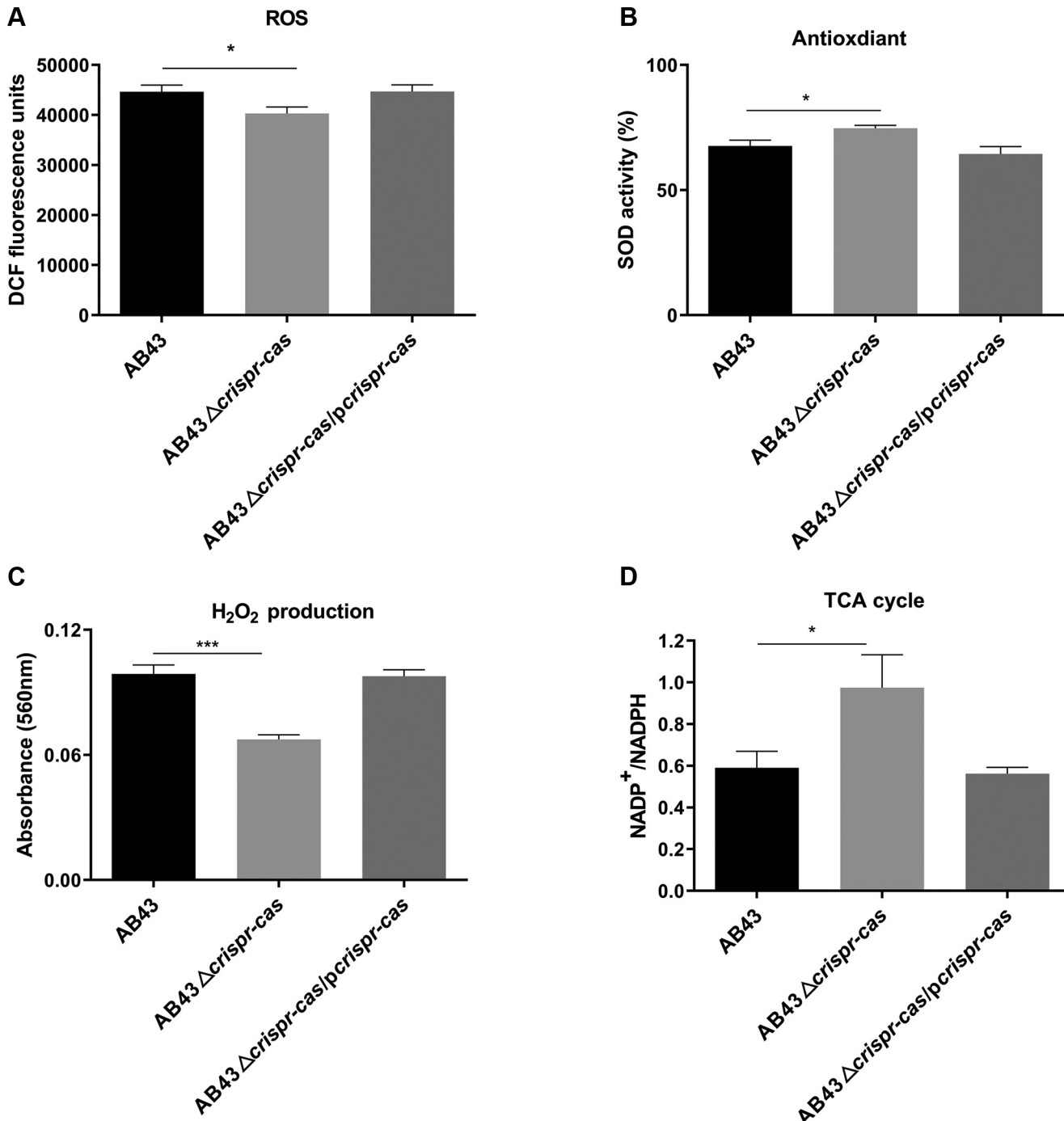

**FIG 5** CRISPR-Cas contributes to reactive oxygen species generation in *A. baumannii*. (A) Deleting *crispr-cas* drastically reduced total ROS. *A. baumannii* strains were probed with 2',7'-dichlorodihydrofluorescein diacetate (DCFH-DA). (B) CRISPR-Cas impairs bacterial oxidative defenses. A biochemical assay measured SOD activity in cells. (C) CRISPR-Cas induces the production of $H_2O_2$. (D) An accelerated TCA cycle was observed in AB43 or complement strains. (A–C) *P*-values (*, $P < 0.05$; ***, $P < 0.001$) were determined by unpaired *t* test between two groups or one-way ANOVA between multiple groups, respectively. All data are presented as the mean $\pm$ SEM.

strains (Fig. 6C). Additionally, *cas3*-negative clinical strains had the highest resistance rates compared with *cas3*-positive in I-Fb strains (Table 1). Previous studies showed that Cas3 could target the QS regulator *lasR* mRNA to reduce bacterial virulence in *P. aeruginosa* (32). Hence, we deduce that *abaI* mRNA might be targeted by crRNA and cleaved by Cas3. From ResFinder (https://cge.cbs.dtu.dk/services/ResFinder/), it is clear

**A**

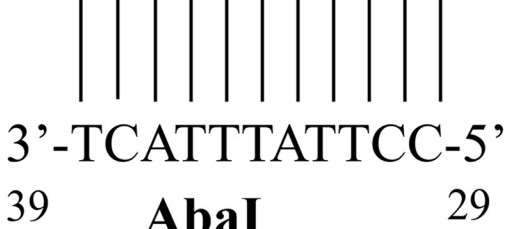

**Spacer 101**

5'-ACTGGAATTCAGCAGTAAATAAGGGGTTAAGA-3'

3'-TCATTTATTCC-5'

39 **AbaI** 29

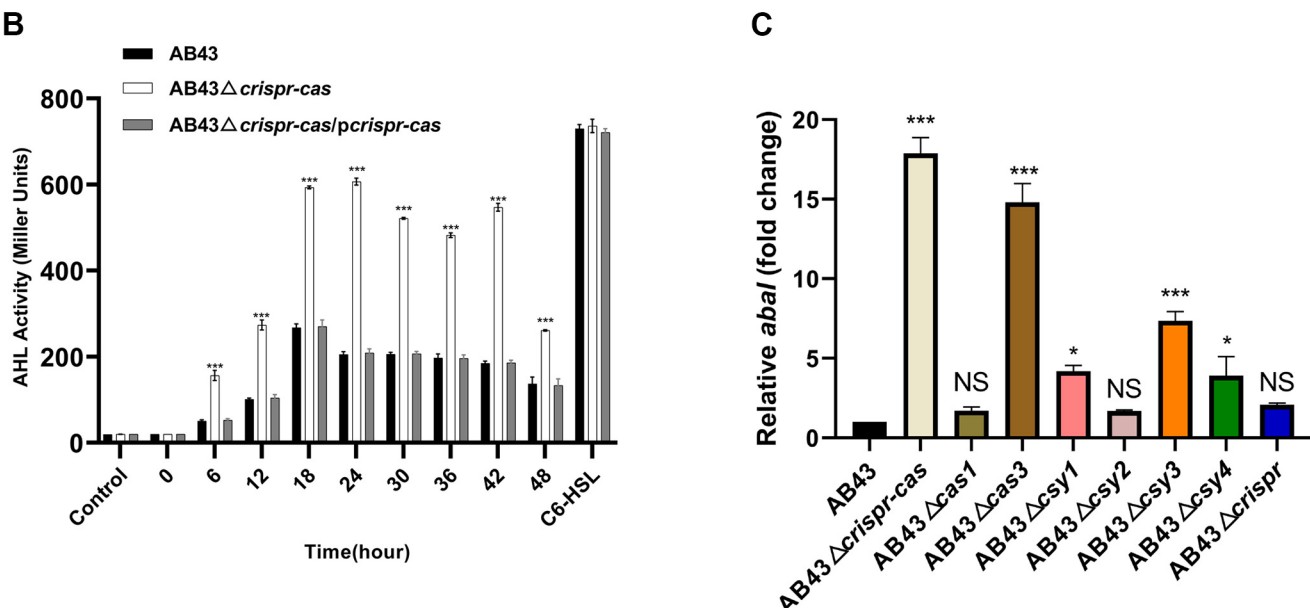

**B**

- AB43
- AB43△*crispr-cas*
- AB43△*crispr-cas*/p*crispr-cas*

**C**

**FIG 6** AB43 crispr-cas deficiency increases the secretion of AHLs. (A) Schematic of predicted hybridization fragments between AbaI and CRISPR. A total of 11 bp of *abaI* mRNA hybridize with spacer 101. (B) AHL production in an entire CRISPR-Cas knockout mutant at different stages of growth in liquid LB medium. AHL in the culture supernatant of *A. baumannii* was quantitated using a $\beta$-galactosidase assay with *A. tumefaciens* KYC55 serving as the reporter strain. A total of 50 $\mu$g/mL of $C_6$HSL was taken as a positive control. (C) *AbaI* transcripts in AB43 and indicated *crispr-cas* deletion mutants were quantified by qRT-PCR. Based on three independent experiments, the results are expressed as the mean $\pm$ SEM ($n$ = 3; one-way ANOVA with Tukey's *post hoc*; \*, $P < 0.05$; \*\*\*, $P < 0.001$; NS, not significant).

that AB43 does not have any acquired resistance genes. It only has the intrinsic *ampC* and *bla*$_{\text{OXA-51-like}}$ genes. Consistent with our inference, *ampC*, and *bla*$_{\text{OXA-51-like}}$ upregulated in all deletion strains except AB43$\Delta$*csy4*, and the AB43$\Delta$*cas3* expression of these two intrinsic drug-resistant genes was highest in all deletion strains (Fig. 8A and B). Similarly, qRT-PCR results showed that most of drug resistance genes: ABC transporters (*macB*, *emrB*) (34), MFS (*craA*, *rpoB*, *tetB*, *abaQ*), the RND superfamily (*adeJ*, *adeB*, *adeG*), the SMR protein family (*abeS*, *abeM*) (35), TCS (*pmrA*, *pmrB*, *adeR*, *adeS*, *bfmS*, *bfmR*, *baeS*) (45), biofilm formation (*ompA*, *ompW*, *lpsB*, *abaI*) (46), clinically significant cephalosporin resistance gene (*ampC*, *bla*$_{\text{OXA-51-like}}$) (47), and other genes (48–51) in the AB43$\Delta$*crispr-cas* and AB43$\Delta$*cas3* were raised (Fig. 8C–J). We also noted that when one specific component of the CRISPR-Cas system was knocked out, the expression of the other components, including *cas3*, were also suppressed (Fig. S5). Taken together, these data indicated that the cleavage activity of Cas3 was the most critical factor in CRISPR-Cas system targeting *abaI* mRNA for degradation.

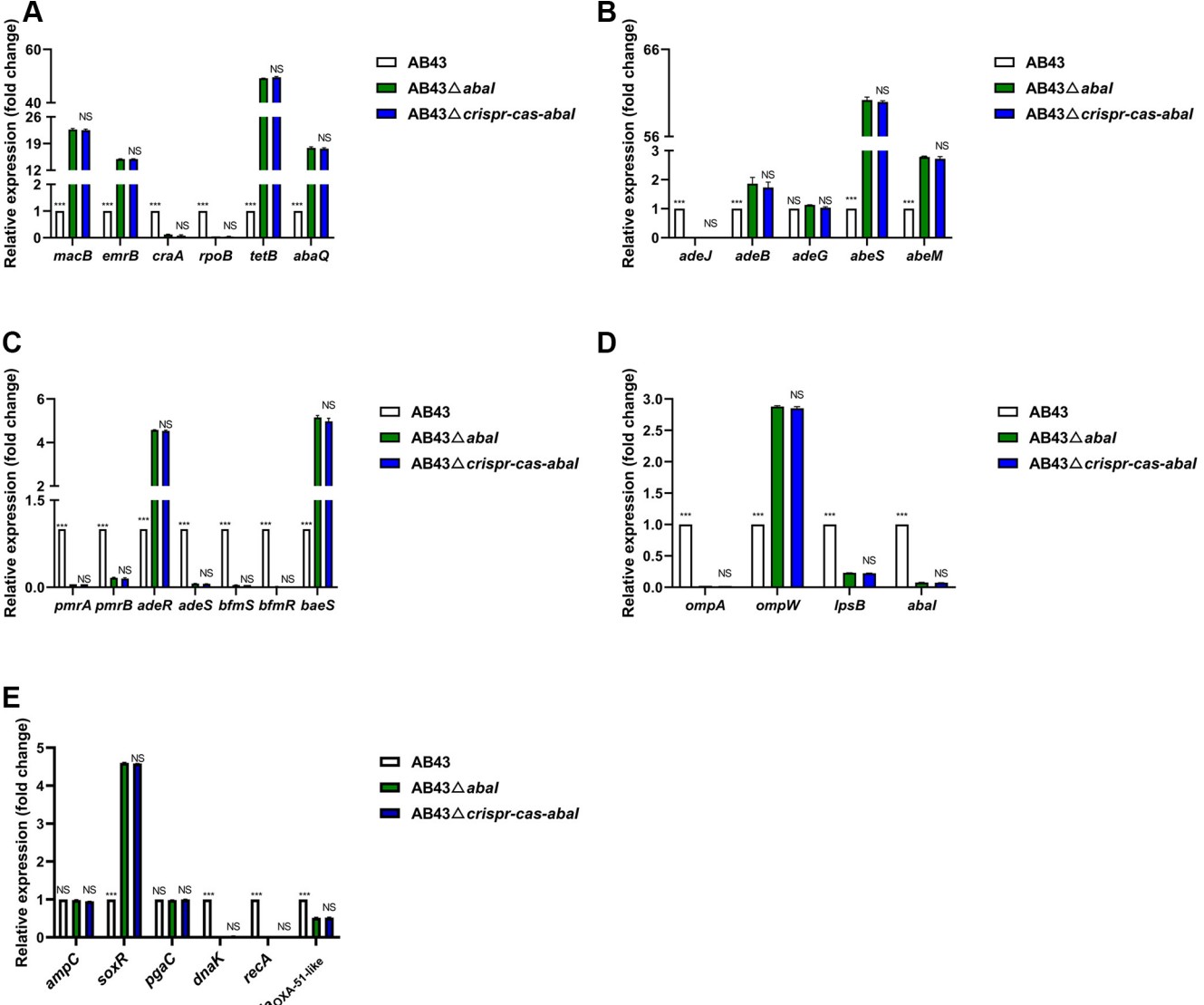

**FIG 7** *AbaI* alters drug-resistant gene expression in AB43. (A–E) RNA was isolated when AB43, AB43Δ*abaI*, and AB43Δ*crispr-cas-abaI* grew to an $OD_{600}$ of 1.0 in liquid LB medium. Transcripts of indicated drug-resistant factors (A) ABC and MFS; (B) RND and SMR; (C) two-component system; (D) biofilm formation; and (E) other clinically relevant antibiotic-resistant genes in AB43-derived strains were quantified by qRT-PCR. This data represents the mean ± SEM from three independent experiments (two-way ANOVA with Tukey's *post hoc*; *, $P < 0.05$; **, $P < 0.01$; ***, $P < 0.001$; NS, not significant).

## DISCUSSION

CRISPR-Cas system is a kind of bacterial immune system that enables bacteria to protect themselves against invasive mobile genetic elements which may carry the genes for antimicrobial resistance (AMR) (12). Many recent studies have shown that the CRISPR-Cas system is closely related to bacterial drug resistance (25, 52, 53). Our previous study observed *cas* gene expression changes in *csy1* single knockout strains under antibiotic stress (27). However, how the CRISPR-Cas systems regulate antibiotic resistance in *A. baumannii* remains unknown. This study focused on a clinical strain AB43, using homologous recombination and sequentially knocking out each component and the entire CRISPR-Cas system to determine their effects on drug resistance. The most interesting finding was that deleting any component of the CRISPR-Cas system rendered AB43 significantly resistant to most of the tested drugs. Furthermore, we demonstrate that the I-Fb CRISPR-Cas system may target and degrade the *abaI* (QS synthase) mRNA, leading to drug resistance-related biological traits and genes being

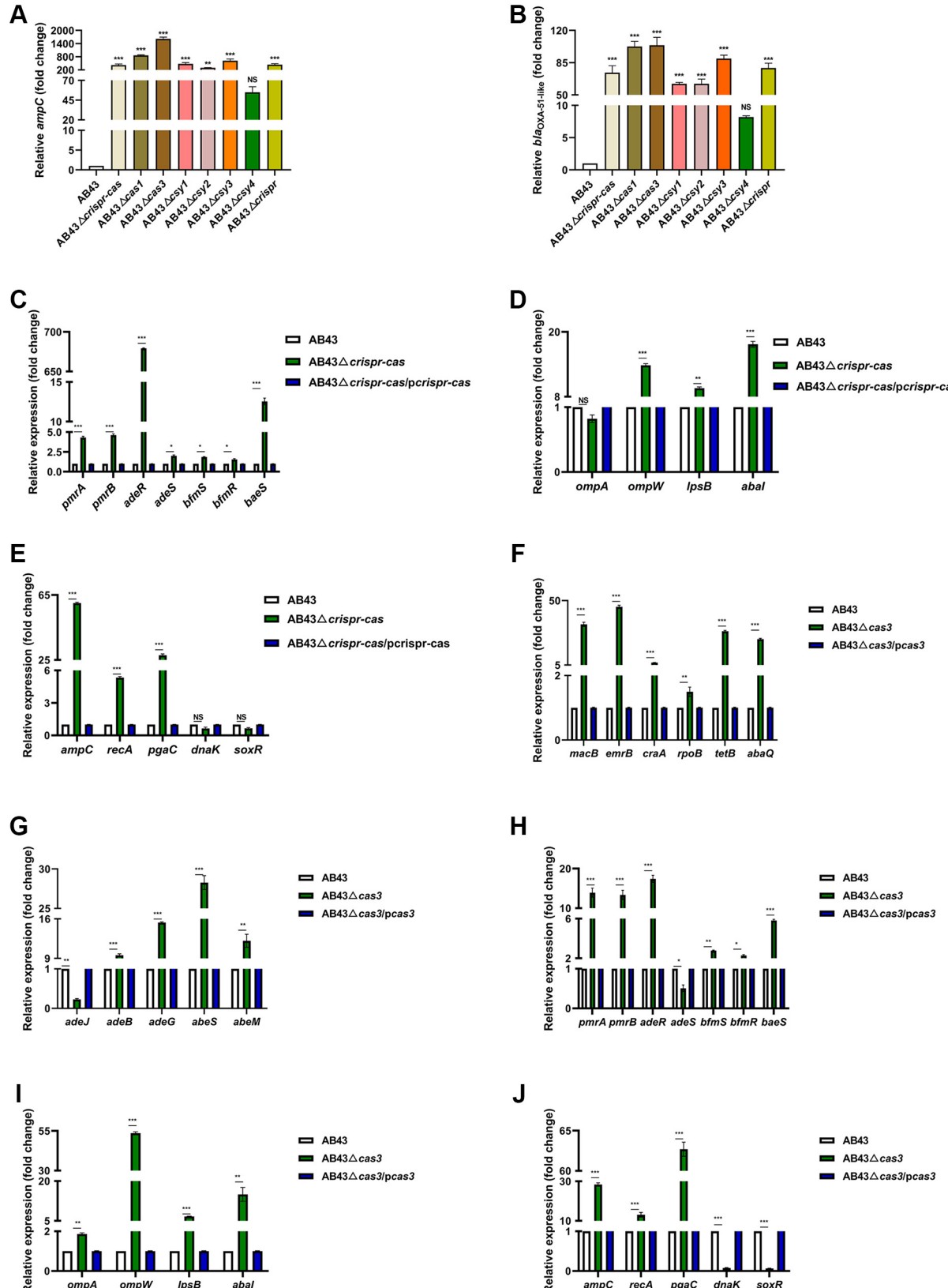

**FIG 8** *Cas3* is necessary for *abaI* mRNA repression in *A. baumannii*. (A and B) mRNA expression of *ampC* and *bla*<sub>OXA-51-like</sub> in different deletion strains. RNA was isolated when AB43, a total CRISPR-Cas cluster deletion mutant, and single element deletion mutants, grew at an OD<sub>600</sub> of 1.0 in liquid LB medium, and transcripts for indicated genes were quantified by qRT-PCR. (C–E) mRNA expression of antibiotic-resistant genes related to (C) two-component systems; (D) biofilm formation; and (E) other clinically relevant antibiotic-resistant genes in

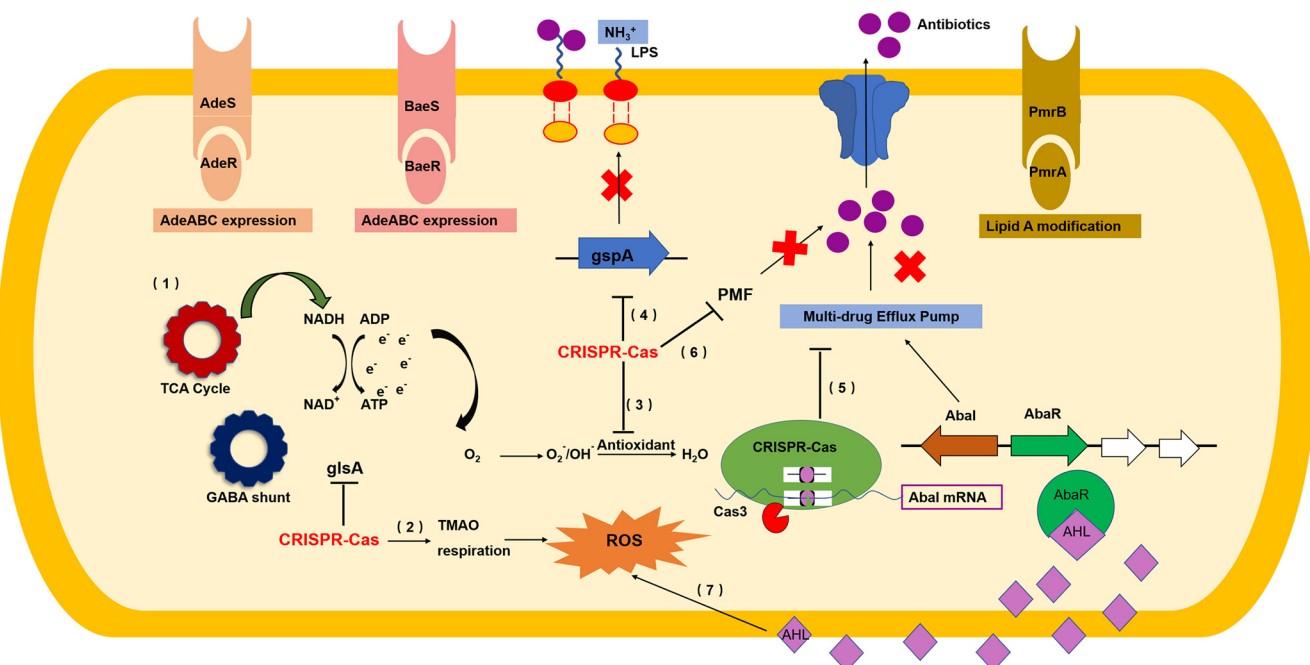

**FIG 9** CRISPR-Cas in *Acinetobacter baumannii* contributes to antibiotic susceptibility by targeting endogenous *abaI* mRNA. Scheme summarizing the mechanisms in which CRISPR-Cas represses antibiotic resistance in *A. baumannii*. First, CRISPR-Cas targets *abaI* to increase ROS. An activated TCA cycle further accelerates the generation of ROS. Additionally, CRISPR-Cas targets *abaI* to suppress the efflux pump while also suppressing the energy required to drive it. CRISPR-Cas targeting *abaI* can also repress lipid A and biofilm formation. Finally, by targeting *abaI*, CRISPR-Cas also controls most multiple drug resistance genes.

inhibited because of the low level of AHLs (Fig. 9). Moreover, Cas3 cleavage activity was the most critical factor in regulating *abaI* mRNA degradation.

The entire CRISPR-Cas represses antibiotic resistance in *A. baumannii*. Of the 245 *A. baumannii* clinical isolates, only two isolates were detected with a complete CRISPR-Cas system, and they were resistant to one category of antibiotics tested. All *cas* gene-negative strains had significantly higher resistance rates than positive strains. This inverse relationship between CRISPR-Cas and antibiotic resistance has been reported in *Enterococcus faecalis* and *Klebsiella pneumoniae* (54, 55). Nonetheless, there were still eight *A. baumannii* with incomplete I-Fb CRISPR-Cas systems resistant to one or two category antibiotics. Compared with AB43, the *cas* gene expression level of these eight strains was significantly increased. It is possible that when a certain Cas component of the I-Fb type is lost, other Cas may have an additional domain that can compensate for the role of a lost Cas (56, 57).

Additionally, most CRISPR-Cas positive strains have incomplete CRISPR-Cas systems. This may be because *A. baumannii* only selectively lost the *cas* gene cluster to adapt to environments with high concentrations of antibiotics, while the remaining incomplete CRISPR-Cas system could not give full play to its immune defense function. Other studies have shown that the CRISPR-Cas system can also be transferred among various bacterial genomes by inserting transposable elements (58), which can explain why some *A. baumannii* have incomplete CRISPR-Cas systems.

Cas3 is necessary for CRISPR-Cas targets *abaI* mRNA to repress antibiotic resistance. We found that with the AB43Δ*cas3* expression of *abaI*, two intrinsic genes *ampC* and *bla*$_{OXA-51-like}$ were highest among the single cas gene knockdown strains.

**FIG 8** Legend (Continued)

AB43Δ*crispr-cas*. (F–J) mRNA expression of antibiotic-resistant genes (F) ABC and MFS; (G) RND and SMR; (H) two-component system; (I) biofilm formation; and (J) other clinically relevant antibiotic resistance genes in AB43Δ*cas3*. This data represents the mean ± SEM from three independent experiments (one-way ANOVA with Tukey's *post hoc*; *, $P < 0.05$; **, $P < 0.01$; ***, $P < 0.001$; NS, not significant).

Additionally, *cas3*-negative clinical strains had higher resistance rates than *cas3*-positive in I-Fb strains. By analogy with that result, Cui et al.'s results suggest that Cas3 targets and downregulates the expression of the operon related to QS, resulting in the expression of *Salmonella* biofilm forming-related genes being changed (59). The Cas3 of *P. aeruginosa* recognizes and cleaves the mRNA for the bacterial QS regulator LasR to dampen the recognition of toll-like receptor 4 (TLR4) (32). Another interesting phenomenon was that when one specific component of the CRISPR-Cas system was knocked out, the expression of *cas3* was suppressed. This further confirms our speculation that loss of either component will result in a decreased expression of *cas3* and eventually lead to drug resistance. A similar phenomenon was found in 4,893 strains of *A. baumannii* collected in the NCBI (60).

CRISPR-Cas represses multidrug efflux pumps by targeting *abaI*. Our results showed that, in AB43Δ*crispr-cas*, EtBr efflux, efflux pump-related gene expression, and energy required for efflux pumps were significantly raised. While in AB43Δ*abaI*, EtBr efflux and energy required for efflux pumps was significantly reduced. Moreover, the expression of several efflux pump-related genes were also reduced in AB43Δ*abaI* and AB43Δ*crispr-cas-abaI*. The positive association between QS-controlled efflux pumps has been commonly identified in many bacterial pathogens. For instance, in *Bacteroides fragilis*, a self-inducing molecular receptor of the QS system reacts to exogenous AHL, upregulating the *bmeB* efflux pump's expression and developing antibiotic resistance (61). An autoinducer can upregulate the multidrug resistance pump MexAB-OprM, allowing bacteria to acquire multidrug resistance in *P. aeruginosa* (62).

At least two TCSs in *A. baumannii* have been reported to upregulate multidrug resistance efflux pumps. AdeRS activates the AdeABC multidrug resistance pump, while BaeSR activates AdeABC, AdeIJK, and MacAB-TolC (7, 63–67). The qRT-PCR results demonstrated that in AB43Δ*abaI* and AB43Δ*crispr-cas-abaI*, *adeR*, *baeS*, and some efflux pump-related genes such as *adeB* are increased. The results were not as expected. Additionally, the expression of *adeB* is increased by about 5-fold in the AB43Δ*crispr-cas* mutant. The results of these two studies seem to be contradictory. This is probably because although AB43Δ*abaI* and AB43Δ*crispr-cas-abaI* were designed to be AHL-deficient, subsequent experiments with these mutants may have been influenced by the presence of *abaI* homologs; AbaI is similar to the LuxI family of autoinducer synthases (68). In addition, it is indicated that a combination of multiple genes may produce efflux pump phenotypes.

Biofilm formation and membrane permeability might also be a specific physiological aspect of CRISPR-Cas targeting *abaI*. AB43Δ*crispr-cas* formed a significantly more robust biofilm; the biofilm-related gene expression and lower membrane permeabilities were induced. In contrast, the biofilm and its related gene expression were significantly reduced in the AB43Δ*abaI* and AB43Δ*crispr-cas-abaI* strains. Biofilms can reduce the sensitivity to aminoglycosides (69), tetracycline, and macrolides by reducing the electrochemical gradient (70). Additionally, QS signals and the resulting downstream consequences can elicit physiological changes that alter the antimicrobial susceptibility of cells within a biofilm. Consistent with this, Niu et al. deleted the *abaI* in *A. baumannii* and found that biofilm formation was dramatically decreased in the mutant strain (71). In *Streptococcus*, competent stimulating factors can enhance the biofilm regulation by the QS system (72). As a regulator of *Staphylococcus epidermidis* biofilm formation, the QS system-related comprehensive regulator *sarA* plays an important role (73, 74).

Inhibition of the CRISPR-Cas system would affect AbaI-AHL complex formation, which might be responsible for a significant induction in catalase and SOD activities. This study found that AB43Δ*crispr-cas* had reduced generation of total ROS and increased SOD activity compared with AB43, while AB43Δ*abaI* had the opposite result. In *A. baumannii* ATCC 17978, inactivation of SOD decreased resistance to oxidative stress and susceptibility to antibiotics (75). QS also regulates the production of 2-n-heptyl-4-hydroxyquinoline-N-oxide (HQNO), a respiratory chain inhibitor that binds to the cytochrome $bc_1$ complex. This causes an accumulation of ROS, a decrease in

membrane potential, and ultimately, autolysis. Autolysis of bacterial cells leads to the release and accumulation of extracellular DNA, enhancing biofilm formation and conferring resistance to positively charged antibiotics (76).

CRISPR-Cas also represses drug-resistant related genes by targeting *abaI*. $\beta$-lactamase is an effective resistance mechanism of *A. baumannii* that can inactivate $\beta$-lactam antibiotics (1). Based on the sequence homology, $\beta$-lactamases were classified into four types: class A extended-spectrum $\beta$-lactamases (ESBL), class B Metallo-$\beta$-lactamases (MBL), class C $\beta$-lactamases (AmpC) and Class D $\beta$-lactamases (OXA) (77). All four types of $\beta$-lactamases were reported in *A. baumannii* (78). Compared to AB43, various resistance genes were expressed at elevated levels in AB43$\Delta$*crispr-cas*. It is striking that the intrinsic drug resistance gene *ampC*, often found in *A. baumannii* from China (79, 80), can be increased up to 200-fold. Another intrinsic drug resistance gene, *bla*$_{OXA-51-like}$ was found to be elevated up to 80-fold. Nevertheless, the expression of *ampC* did not change and *bla*$_{OXA-51-like}$ significant reduction in AB43$\Delta$*abaI* and AB43$\Delta$*crispr-cas-abaI*. A similar phenomenon has been previously reported. According to Dou et al., AHLs generated by *A. baumannii* might increase the expression of drug-resistance genes such as *bla*$_{OXA-51-like}$, *ampC*, *adeA*, and *adeB* (81).

This study revealed that CRISPR-Cas3 in *A. baumannii* contributes to antibiotic susceptibility by targeting endogenous genes *abaI*. Accordingly, this knowledge provides a new way to understand the functions of CRISPR-Cas systems in the drug sensitivity of *A. baumannii*. Our findings also provide vital information for the correlation study of CRISPR-Cas systems and *A. baumannii* drug resistance. However, the exact mechanism of how the CRISPR-Cas system targets *abaI* is not yet entirely distinct. It remains to be elucidated in future work whether CRISPR-Cas systems confer a similar susceptibility to antibiotics in other bacterial species as what has been observed in *A. baumannii*.

## MATERIALS AND METHODS

**Bacterial isolates.** A total of 245 clinical *A. baumannii* strains were randomly collected from five hospitals. These five sources and their isolates were as follows: 97 isolates from the Affiliated Hospital of Yangzhou University, harvested between 2017 to 2019; 84 isolates from the Affiliated Zhangjiagang Hospital of Soochow University, harvested between 2019 to 2020; 27 isolates from Northern Jiangsu People's Hospital, harvested between 2017 to 2018; 21 isolates from Xuyi People's Hospital, harvested between January and April 2019; and, 16 isolates from Wuxi People's Hospital, harvested between June and December 2018. The 245 clinical *A. baumannii* strains were isolated from sputum ($n = 181$, 73.81%), urine ($n = 17$, 7.14%), shunt fluids ($n = 17$, 7.14%), blood ($n = 15$, 5.95%), and other sources ($n = 15$, 5.96%). All bacterial strains were grown at 37°C on Luria-Bertani (LB) agar plates and subjected to identification by API20 NE (bioMérieux Marcy-l'Étoile, France) (82).

**Detection of CRISPR-Cas systems.** PCR determined the prevalence of CRISPR-Cas systems. The DNA of all bacterial isolates were extracted by boiling using $1\times$ TE solution (10 mmol/L Tris-HCl and 1 mmol/L EDTA; Sangon, China). PCR primers were designed to detect the CRISPR-Cas system using *A. baumannii* ATCC 19606 and AB43 as reference strains, listed in Table S1 (Supporting Information). PCR cycle was as follows: predenaturation at 95°C for 5 min, denaturation at 94°C for 1 min, annealing at the optimal temperature of different primers for 30 s, and extension at 72°C for 1 min. Denaturation, annealing, and extension steps were repeated for 35 cycles, with a final extension step at 72°C for 10 min. PCR products were separated and detected by gel electrophoresis and were confirmed by DNA sequencing (Sangon, China) and nucleotide BLAST search using GenBank.

**Multilocus sequence typing (MLST).** MLST was performed according to a previously described study (83). Briefly, internal fragments of seven housekeeping genes (*gltA*, *gyrB*, *gdhB*, *recA*, *cpn60*, *gpi*, and *rpoD*) were amplified by PCR (84). Primers are listed in Table S2 (Supporting information). The sequences of these seven housekeeping genes were compared with existing sequences in the MLST database (http://pubmlst .org/abaumannii/) to assign allelic numbers. Sequence types (STs) were assigned according to their allelic profiles.

**Antimicrobial susceptibility testing.** Based on Clinical and Laboratory Standards Institute (CLSI) guidelines (CLSI, 2016), antibiotic resistance was assessed on Mueller-Hinton agar using the disk diffusion or broth microdilution method. All *A. baumannii* clinical isolates were tested for susceptibility to nine types of antimicrobial agents (penicillin, $\beta$-lactam/$\beta$-lactamase inhibitor combinations, cephems, carbapenems, aminoglycosides, fluoroquinolones, lipopeptides, tetracyclines, and folate pathway inhibitors). ATCC 19606 and ATCC17978 were used as controls. Isolates resistant to three or more classes of antibiotics were classified as multidrug-resistant.

**Construction of AB43 deletion mutants and complemented strains.** As previously described, in-frame deletion mutants of CRISPR-Cas genes were made using a recombineering system for targeted genome editing (85). Briefly, using AB43 genomic DNA and PKD4 as templates, the upstream and downstream homology arms of the target fragment and the kanamycin cassette fragment with FRT

site were amplified, respectively. Three PCR amplicons containing overlapping regions were assembled using overlap extension PCR with specific primers (Table S4), and the resulting fragment was electroporated into competent AB43 carrying pAT04, which expresses the RecAB recombinase. Transformants were selected on LB plates containing 7.5 $\mu$g/mL kanamycin, and PCR confirmed integration of the resistance marker. To remove the kanamycin resistance cassette, electrocompetent mutants were transformed with pAT03 plasmid, which expresses the FLP recombinase. A loss of kanamycin resistance was observed in these colonies, confirmed by PCR, and sequenced using identification primers (Table S4).

Complementation vectors for the deletion strains were constructed using the primer sets in Table S4. The full-length genes with their native promoters were amplified, cloned into vector PABBR-MCS (85), and electroporated into corresponding mutant strains.

**In vitro growth kinetics sssays.** Overnight AB43 cultures of all strains were prepared and equalized by dilution adjustments to assess the growth of AB43 strains in LB. Optical density at 600 nm ($OD_{600}$) was used to measure the cell densities.

**Crystal violet staining.** We measured the biofilm formation of *A. baumannii* by staining with crystal violet (86). Overnight cultures of *A. baumannii* strains were diluted 1:100 in fresh liquid LB and grown to $OD_{600}$ = 0.5 ($5 \times 10^8$ CFU/mL). Next, *A. baumannii* strains were cultured in sterile 96-well microtiter plates for 24 h at 37°C. After incubation, the supernatant planktonic bacteria were removed, and the wells were washed three times with phosphate-buffered saline (PBS). The biofilm was fixed with 4% paraformaldehyde and then washed again. The residual biofilms were then stained with 200 $\mu$L of 0.1% (1 g/L) crystal violet and incubated for 10 min with gentle agitation. The plates were washed twice with PBS, and 200 $\mu$L of 30% acetic acid was added to solubilize the dye. Finally, the acetic acid was transferred to a new plate, and the absorbance at 570 nm (BioTek, USA) was recorded (86). This experiment performed three technical replicates, and averaged the results. LB broth uninoculated with bacteria was used as a negative control.

**Cell membrane permeability assessment.** Overnight cultures of *A. baumannii* strains were adjusted to an $OD_{600}$ of approximately 0.5. Next, 10 nM propidium iodide (PI) was added to all samples, and the suspensions were incubated at 37°C for 20 min. After washing three times with PBS, we used an excitation wavelength of 535 nm and an emission wavelength of 615 nm to measure the fluorescence intensity of 10 nM PI-labeled cells in *A. baumannii* strains (87).

**Outer membrane permeability assessment.** The fluorescent probe N-Phenyl-1-naphthylamine (NPN) (10 $\mu$M) (88) was used to evaluate the outer membrane permeability of *A. baumannii* strains. Overnight cultures were harvested by centrifugation, washing, and resuspension in PBS to $OD_{600}$ = 0.5. The dye NPN was added to a final concentration of 10 $\mu$mol/L. Cells were then incubated at 37°C for 30 min, and the fluorescence was measured using a Tecan Infinite M200 Microplate Reader. The emission wavelength was 350 nm, and the excitation wavelength was 420 nm.

**Efflux pump assays.** An EtBr efflux assay was performed based on a previous study to test the effect of CRISPR-Cas on the inhibition of efflux pumps (89). Strains were resuspended with PBS to an $OD_{600}$ of 0.5. Cells were then coincubated with EtBr (8 $\mu$g/mL final concentration) or the known efflux pump inhibitor CCCP ($10^{-4}$ M) at 37°C. We then centrifuged the pellets at 5,000 g for 10 min at 4°C and resuspended these pellets in fresh liquid LB. We measured the EtBr efflux in cells for 60 min using a 530-nm excitation wavelength and a 600-nm emission wavelength.

**Membrane depolarization assays.** 3,3′-Dipropylthiadicarbocyanine iodide (DiSC3(5), 0.5 $\mu$M) was utilized to determine membrane potentials (90). *A. baumannii* bacterial cells were washed and resuspended to obtain an $OD_{600}$ of 0.5 with PBS. DiSC3(5) was added (Aladdin, catalogue number D131315; ≥98%, 0.5 $\mu$mol/L) to the mixture. The dissipated membrane potential of *A. baumannii* strains was measured with an excitation wavelength of 622 nm and an emission wavelength of 670 nm.

**Proton motive force assessment.** The proton motive force of *A. baumannii* strains was measured with the pH-sensitive fluorescence probe BCECF-AM ($2 \times 10^{-5}$ M). After 30 min of incubation at 37°C, the fluorescence spectrometer's excitation and emission wavelengths were set to 500 and 522 nm, respectively, to monitor the proton motive force of *A. baumannii* strains (91).

**ATP level determination.** Intracellular ATP levels of *A. baumannii* strains were determined using an Enhanced ATP assay kit (Beyotime, China). Overnight cultures of strains were resuspended to obtain an $OD_{600}$ of 0.5 with 0.01 mol/L PBS (pH 7.4). Then the bacterial precipitates were lysed by lysozyme and centrifuged to be used to measure intracellular ATP levels. A 96-well plate containing the detecting solution was incubated at room temperature for 5 min. Supernatants were added and mixed immediately, and an Infinite M200 Microplate reader (Tecan) was used to monitor luminescence signals (92).

**Total ROS and H$_2$O$_2$ measurement.** As previously described (37), 2′,7′-dichlorodihydrofluorescein diacetate (DCFH-DA, 10 $\mu$M) was applied to monitor the levels of ROS in *A. baumannii* strains following the manufacturer's instructions (Beyotime, China). After incubation at 37°C for 30 min, 200 $\mu$L of probe-labeled bacterial cells was added to a 96-well plate, and fluorescence units were immediately measured with an excitation wavelength of 488 nm and emission wavelength of 525 nm using an Infinite M200 Microplate reader (Tecan). The production of $H_2O_2$ in *A. baumannii* strains was assessed using a Hydrogen Peroxide assay kit (Beyotime, China). The absorbance of lysis buffer at 560 nm was measured after 1 h of incubation.

**SOD activity assessment.** Intracellular superoxide dismutase (SOD) activity in *A. baumannii* strains was measured using the Total Superoxide Dismutase assay kit with WST-8 (S0101, Beyotime, China). Based on the manufacturer's instructions, 450 nm was used to measure absorbance.

**TCA cycle measurements.** The intracellular concentrations of NAD$^+$ and NADH were determined using an enzymatic cycling assay kit (Beyotime, China). *A. baumannii* was collected and diluted to an OD$_{600}$ of 0.5 in PBS and then resuspended in 200 $\mu$L of precooled extraction buffer. The lysate was centrifuged at 12,000 g for 10 min at 4°C, and the absorbance of the supernatant at 450 nm was measured. The concentrations were measured in quadruplicate from three independent experiments.

**RNA isolation and qRT-PCR.** RNA isolation, cDNA synthesis, and PCR amplification were carried out as described previously (93). *A. baumannii* were grown overnight in LB broth and diluted 1:100 into 5 mL fresh LB supplemented with indicated antibiotics. After bacterial cells were grown to an OD$_{600}$ of 1 at 37°C, bacterial cells were harvested for total RNA using the RNAprep pure bacteria kit (TIANGEN, Beijing, China), and total RNA was assessed using the ratio of 260 nm/280 nm absorbance on a Nanodrop spectrophotometer. The RNA extracted from all bacterial cells was adjusted to equal concentrations before cDNA synthesis. According to the manufacturer's protocol, the reverse transcription of 1 $\mu$g extracted RNA was performed using a PrimeScript RT reagent kit with gDNA Eraser (TaKaRa, Beijing, China), and 10 ng of cDNA was used as a template for qRT-PCR. All primers used in this study for qRT-PCR are listed in Table S12 (Supporting information). Three independent qRT-PCRs were performed using the 7500 Fast real-time PCR system (Applied Biosystems, CA, USA). SYBR green I fluorescence in every cycle was monitored by the system software, and the threshold cycle (CT) was determined. 16S rRNA was used as an internal control, and the relative gene expression level was calculated using the $2^{-\Delta\Delta Ct}$ method (94).

**Transcriptomic analysis.** Strains were grown in liquid LB to the early exponential phase. We extracted total RNA from these samples using an EASY Spin Plus kit (Qiagen, Hilden, Germany). We quantified RNA based on the ratio of absorbance (260 nm/280 nm) using a Nanodrop spectrophotometer (Thermo Scientific) and sequenced samples by using the Illumina Hiseq system (Sangon, Shanghai, China). According to the manufacturer's protocol, the library construction of purified mRNA was conducted using an Illumina Truseq RNA sample prep kit. The raw read counts were normalized with DESeq2 (95) to estimate gene expression and identify differentially expressed genes. Differential gene expression was identified using a threshold of $P < 0.05$, a fold change of at least two, and a false discovery rate (FDR) $< 0.2$. Differences between two *A. baumannii* strains were analyzed using Cuffdiff software (http://cufflinks.cbcb.umd.edu/).

**N-Acyl homoserine lactone (AHL) quantification by $\beta$-galactosidase assay.** AHL levels were quantified in cell-free culture media taken from AB43 strains at various times using a method described by previous work (96), which involves the use of the $\beta$-galactosidase reporter strain *A. tumefaciens* KYC55 (68). The culture supernatant was obtained by centrifugation, and it was extracted three times with equal amounts of acidified ethyl acetate (0.1% vol/vol glacial acetic acid). We dried the extracts in a fume hood, resuspended them in 1 mL of acidified ethyl acetate, and then dried them again in a fume hood. In order to rehydrate the dried extracts, 1 mL of acetonitrile was used. The positive control was 50 $\mu$L of synthesized N-hexanoyl-L-homoserine lactone (C6-HSL) at 50 $\mu$g/mL, and the negative control was blank LB media. All experiments were repeated at least three times.

**Statistical analyses.** Statistical analysis was performed using GraphPad Prism 8 and SPSS software. All data are presented as the mean $\pm$ SEM. The endpoint differences for growth curves (OD$_{600}$) were analyzed using a two-tailed paired *t* test. Unless otherwise noted, an unpaired *t* test between two groups or one-way ANOVA between multiple groups was used to calculate *P*-values (*, $P < 0.05$; **, $P < 0.01$; ***, $P < 0.001$).

**Data analysis.** The whole genome sequence of *A. baumannii* strain AB43 was submitted to GenBank under accession number CP083181. The transcriptome sequencing data described was submitted to the NCBI Sequence Read Archive under accession number SRR17299399.

## SUPPLEMENTAL MATERIAL

Supplemental material is available online only.
**SUPPLEMENTAL FILE 1**, PDF file, 2.1 MB.

## ACKNOWLEDGMENTS

We thank Bryan W. Davies from University of Texas at Austin and Yajun Song from the Beijing Institute of Microbiology and Epidemiology Academy of Military Medical Sciences for providing plasmids to construct the deletion and complemented mutants. We thank Hui Wang (Nanjing Agricultural University, China) for the kind donation of the biosensor strain *A. tumefaciens* KYC55.

This work was supported by the National Natural Science Foundation of China (82073611, 82002186).

We declare that this research was conducted without any commercial or financial relationships construed as a potential conflict of interest.

Y.W. and G.L. designed the experiments; Y.W., J.Y., X.S. and M.L. conducted the experiments; Y.W., J.Y., X.S., P.Z., and Z.Z. analyzed the data, prepared the figures and tables, and drafted the manuscript; Y.W., T.G., H.J., and G.L. revised and approved the final manuscript.

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
