## [Reviewer comments · Microbiology Spectrum]

Microbiology Spectrum

CRISPR-Cas in *Acinetobacter baumannii* Contributes to Antibiotic Susceptibility by Targeting Endogenous *Abal*

Yuhang Wang, Jie Yang, Xiaoli Sun, Mengying Li, Pengyu Zhang, Zhongtian Zhu, Hongmei Jiao, Tingting Guo, and Guocai Li

Corresponding Author(s): Guocai Li, Yangzhou University

Review Timeline:

Submission Date:	March 4, 2022
Editorial Decision:	May 9, 2022
Revision Received:	July 12, 2022
Accepted:	July 14, 2022

Editor: Monica Garcia-Solache

Reviewer(s): Disclosure of reviewer identity is with reference to reviewer comments included in decision letter(s). The following individuals involved in review of your submission have agreed to reveal their identity: Jiaqi Cheng (Reviewer #1)

Transaction Report:

DOI: <https://doi.org/10.1128/spectrum.00829-22>

May 9, 2022

Dr. Guocai Li
Yangzhou University
Pathogen Biology and Immunology
11 Huai-hai Road
Yangzhou, Jiangsu 225001
China

Re: Spectrum00829-22 (CRISPR-Cas3 in *Acinetobacter baumannii* Contributes to Antibiotic Susceptibility by Repressing Efflux Pumps in a Quorum-Sensing Dependent Manner)

Dear Dr. Guocai Li:

Link Not Available

Sincerely,

Monica Garcia-Solache

Journals Department
Reviewer comments:

Reviewer #1 (Comments for the Author):

General comments:

This is a well-written paper. The hypothesis building is very solid and make sense. The data is very well-rounded and favorably tested the hypothesis. The concept and findings are of interest, but many opportunities exist to improve the clarity of the presentation. Especially compared to the considerable amount of data, the figure legend and the result part contain not enough information to support readers to go through those data and follow the hypothesis building.

Specific comments:

.The importance section is a little bit redundant, it can be cut shorter or deleted since the content is somehow covered in abstract and discussion.

.Line 63-89, the introduction of CRISPR-Cas is not in a very good order and may lead to confusion. Try starting with overall introduction of CRISPR-Cas in bacteria immune system, then lead to I-Fb CRISPR-CAS and I-Fb CRISPR-CAS3

.Line 91, the majority of strains do not have the presence of CRISPR-Cas, so this title might be mislead.

.Line 95, define MDR (resistant to three or more classes of antibiotic)and sensitive.

.line 112, if you want to show no differences between group, an endpoint statistical test is needed.

.Line 114, the supplement table S3 which should support the conclusion is not well-organized and with no table title or description. You may reorganize the table S3 or only make conclusion draw from table 1.

.line 139 Can't draw conclusion about drug resistance here, only resistance gene expression level

.line 160 figure 4B and 4C clarify which genes are related to efflux pump and which are related to TCS

.line 235 From figure 7F and 7G, if the ANOVA test of the two figure are done separately, the significancy can not be combined to get 62.5%.

.line 240-244, the explanation for figure 8 is too vague, hard to understand what those gene in 8E-8L are. This paragraph is too general.

.line 258 There is no figure 9.

.line396 Is there a washing step for permeability assessment?

.Line 765 I represents?

.Line 838 Why separate 7F and 7G? Was the ANOVA test done separately in 7F and 7G?

Reviewer #2 (Public repository details (Required)):

A whole genome sequence of *A. baumannii* strain AB43 has been submitted to GenBank with the accession number CP083181.

A transcriptome sequencing data has been submitted to NCBI with the accession number SRR17299399.

Reviewer #2 (Comments for the Author):

In this manuscript, authors verified the function of CRISPR-Cas system in rendering *A. baumannii* antibiotic sensitivity through various ways, such as inhibiting hydrolyzing enzymes, efflux pumps and biofilm formation, production of ROS and H₂O₂, decreasing the activity of SOD and NADP. However, the reason of the above changes is concluded to the regulation axis of Cas3 - Abal - efflux pumps. It seems not very persuasive that there is not enough evidence to prove that Cas3 is the core of CRISPR-Cas system to lead to the outcomes, since the deletion of other components in this system could also affect the expression of Cas3 and bring similar results. Moreover, changes of Abal and efflux pumps are not the only reason for antibiotic sensitivity of *A. baumannii*. So, the function of CRISPR-Cas is not clearly explained by the current results.

General comments:

The manuscript is difficult to follow, many of the sentences need revision, and English needs an extensive proofreading.

The results section includes a discussion of results, while the discussion section reports a repetition of results. In addition, the discussion section does not fully explain all the results obtained in this work. Both sections need to be revised.

There is a common problem in the manuscript including figures and tables, that gene names should be all lowercase in italic, and protein names should be capitalized for the first letter and non-italic. Take Abal as an example, its gene name is *abal*, and protein name is Abal. Misused of the names are all through the manuscript.

Specific comments:

In some sentences, the expression is not rigorous. For example, in line 17-20, the factors of altered targets, decreased permeability, overexpression of efflux pumps, metabolic changes and biofilm formation could cause resistance phenotype in *A. baumannii*, but not the way of obtaining multidrug resistance genes. It's better to say "obtains multidrug resistance phenotype".

In line 26 and 65, authors say AB43 has a complete I-Fb CRISPR-Cas system. Please clarify what is the type of I-Fb, since according to the references, the type I CRISPR-Cas system are classified as Type I-F1, Type I-F2 (Ref. 25). Please use the official classification name and provide correct references.

In line 79-80, the authors describe that QS activates type I-F CRISPR-Cas expression and CRISPR adaptation in *P. aeruginosa*. And in line 84-85, authors say that the purpose of this work is to investigate the role of the Type I-Fb CRISPR-Cas system in modulating QS operation in *A. baumannii*. What's the relationship between QS and Type I-Fb CRISPR-Cas system? Who regulates who? Or are they in a regulation circle? There should be some background about this knowledge.

Between line 91-101, please list the analysis of antibiotic susceptibility results of all the 245 clinical strains. The

number/percentage of MDR and sensitive strains in 132 CRISPR-Cas negative isolates should also be shown in the results.

Line 98-101, should it be related between MDR and missing of component of the CRISPR-Cas system? Since most of the strains with incomplete or without CRISPR-Cas system are MDR.

In Figure 1B, the subscripts of X axis are wrong, they should be I-Fa and I-Fb. Please also check whether they should be I-F1 and I-F2.

In line 117-118, authors say that WT and all the gene rescue mutants were susceptible to all the 24 antibiotics. However, in Table 1, AB43 and the complement strain are resistant to piperacillin, and intermediate to cefotaxime and ceftriaxone. Please correct the results.

Line 124, blaOXA-51-like should be blaOXA-51-like. Moreover, as shown in the results that the expression of ampC and blaOXA-51-like were increased significantly in the Δ CRISPR-Cas mutant, why don't authors consider the inhibition function of CRISPR-Cas on these two enzymes, but give the conclusion only on regulation of efflux pumps?

Line 193-195, the explanation sounds conflict with the results. Since dysfunction of the CRISPR-Cas system could enhance biofilm formation, and at the same time dampen bacterial permeability, both two results will result in the increased resistance to antibiotics, thus show a synergistic effect with efflux pumps, but not limit the efflux of antibiotics.

Line 240, authors say that several drug resistance factors are under that control of Abal according to ref. 47. However, Abal is not mentioned in this reference. BTW, authors did not say which factors are under the control of Abal, thus the expression results could hardly be explained.

In Figure 8F, the abal expression is increased in AB43 Δ CRISPR-Cas mutant, that means CRISPR-Cas system represses the function of Abal. And in Figure 8I, adeB expression is increased in Abal knockout mutant, which shows another negative regulation. In this case, downregulation of CRISPR-Cas could result in an upregulation of abal, and finally lead to a decrease of adeB expression. However, in Figure 4C, the expression of adeB is increased for about 5 folds in Δ CRISPR-Cas mutant. How to explain the conflict results?

Moreover, if the results in Figure 8 are correct, expression of CRISPR-Cas will inhibit the activity of Abal, which will boost the expression of AdeB, but not lead to ABC transporter inhibition as mentioned by authors in line 258.

In addition, in line 257-258, authors describe the relationship among Cas3, Abal and ABC transporter, but all the qPCR results are obtained in Δ CRISPR-Cas mutant, and there is no evidence about the changes of abal and adeB expression in Cas3 knockout mutant.

There are many mistakes in the manuscript, for example, ATCC 19606 is written as ATTC 19606 in line 346; nine types of antimicrobial agents are used in the work, but is written as nine antimicrobial agents in line 363.

Line 367-377, the knockout protocol is not written in the actual way of procedure.

At the first appearance of an abbreviation, the full name should be provided, such as OD. Besides, the description of OD600 is not equal in the manuscript that it is also written as OD600nm and OD600 in some sentences.

Line 402, for the unit of NPN, should it be 10 μ m or 10 μ M?

μ l and μ L are mixed used.

When writing the P value, P should be in italic. Moreover, sometimes it is shown in uppercase, but sometimes in lowercase.

Line 487, " μ g ml⁻¹" is a wrong way of showing the unit.

Staff Comments:

Preparing Revision Guidelines

- Point-by-point responses to the issues raised by the reviewers in a file named "Response to Reviewers," NOT IN YOUR COVER LETTER.

- Upload a compare copy of the manuscript (without figures) as a "Marked-Up Manuscript" file.
- Each figure must be uploaded as a separate file, and any multipanel figures must be assembled into one file.
- Manuscript: A .DOC version of the revised manuscript
- Figures: Editable, high-resolution, individual figure files are required at revision, TIFF or EPS files are preferred

Please return the manuscript within 60 days; if you cannot complete the modification within this time period, please contact me. If you do not wish to modify the manuscript and prefer to submit it to another journal, please notify me of your decision immediately so that the manuscript may be formally withdrawn from consideration by Microbiology Spectrum.

In this manuscript, authors verified the function of CRISPR-Cas system in rendering *A. baumannii* antibiotic sensitivity through various ways, such as inhibiting hydrolyzing enzymes, efflux pumps and biofilm formation, production of ROS and H₂O₂, decreasing the activity of SOD and NADP. However, the reason of the above changes is concluded to the regulation axis of Cas3 – Abal - efflux pumps. It seems not very persuasive that there is not enough evidence to prove that Cas3 is the core of CRISPR-Cas system to lead to the outcomes, since the deletion of other components in this system could also affect the expression of Cas3 and bring similar results. Moreover, changes of Abal and efflux pumps are not the only reason for antibiotic sensitivity of *A. baumannii*. So, the function of CRISPR-Cas is not clearly explained by the current results.

General comments

The manuscript is difficult to follow, many of the sentences need revision, and English needs an extensive proofreading.

The results section includes a discussion of results, while the discussion section reports a repetition of results. In addition, the discussion section does not fully explain all the results obtained in this work. Both sections need to be revised.

There is a common problem in the manuscript including figures and tables, that gene names should be all lowercase in italic, and protein names should be capitalized for the first letter and non-italic. Take Abal as an example, its gene name is *abal*, and protein

name is Abal. Misused of the names are all through the manuscript.

Specific comments

In some sentences, the expression is not rigorous. For example, in line 17-20, the factors of altered targets, decreased permeability, overexpression of efflux pumps, metabolic changes and biofilm formation could cause resistance phenotype in *A. baumannii*, but not the way of obtaining multidrug resistance genes. It's better to say "obtains multidrug resistance phenotype".

In line 26 and 65, authors say AB43 has a complete I-Fb CRISPR-Cas system. Please clarify what is the type of I-Fb, since according to the references, the type I CRISPR-Cas system are classified as Type I-F1, Type I-F2 (Ref. 25). Please use the official classification name and provide correct references.

In line 79-80, the authors describe that QS activates type I-F CRISPR-Cas expression and CRISPR adaptation in *P. aeruginosa*. And in line 84-85, authors say that the purpose of this work is to investigate the role of the Type I-Fb CRISPR-Cas system in modulating QS operation in *A. baumannii*. What's the relationship between QS and Type I-Fb CRISPR-Cas system? Who regulates who? Or are they in a regulation circle? There should be some background about this knowledge.

Between line 91-101, please list the analysis of antibiotic susceptibility results of all the

245 clinical strains. The number/percentage of MDR and sensitive strains in 132 CRISPR-Cas negative isolates should also be shown in the results.

Line 98-101, should it be related between MDR and missing of component of the CRISPR-Cas system? Since most of the strains with incomplete or without CRISPR-Cas system are MDR.

In Figure 1B, the subscripts of X axis are wrong, they should be I-Fa and I-Fb. Please also check whether they should be I-F1 and I-F2.

In line 117-118, authors say that WT and all the gene rescue mutants were susceptible to all the 24 antibiotics. However, in Table 1, AB43 and the complement strain are resistant to piperacillin, and intermediate to cefotaxime and ceftriaxone. Please correct the results.

Line 124, bla_{OXA-51-like} should be *bla*_{OXA-51-like}. Moreover, as shown in the results that the expression of *ampC* and *bla*_{OXA-51-like} were increased significantly in the Δ CRISPR-Cas mutant, why don't authors consider the inhibition function of CRISPR-Cas on these two enzymes, but give the conclusion only on regulation of efflux pumps?

Line 193-195, the explanation sounds conflict with the results. Since dysfunction of the CRISPR-Cas system could enhance biofilm formation, and at the same time dampen bacterial permeability, both two results will result in the increased resistance to antibiotics,

thus show a synergistic effect with efflux pumps, but not limit the efflux of antibiotics.

Line 240, authors say that several drug resistance factors are under that control of Abal according to ref. 47. However, Abal is not mentioned in this reference. BTW, authors did not say which factors are under the control of Abal, thus the expression results could hardly be explained.

In Figure 8F, the *abaI* expression is increased in AB43 Δ CRISPR-Cas mutant, that means CRISPR-Cas system represses the function of Abal. And in Figure 8I, *adeB* expression is increased in Abal knockout mutant, which shows another negative regulation. In this case, downregulation of CRISPR-Cas could result in an upregulation of *abaI*, and finally lead to a decrease of *adeB* expression. However, in Figure 4C, the expression of *adeB* is increased for about 5 folds in Δ CRISPR-Cas mutant. How to explain the conflict results?

Moreover, if the results in Figure 8 are correct, expression of CRISPR-Cas will inhibit the activity of Abal, which will boost the expression of AdeB, but not lead to ABC transporter inhibition as mentioned by authors in line 258.

In addition, in line 257-258, authors describe the relationship among Cas3, Abal and ABC transporter, but all the qPCR results are obtained in Δ CRISPR-Cas mutant, and there is no evidence about the changes of *abaI* and *adeB* expression in Cas3 knockout mutant.

There are many mistakes in the manuscript, for example, ATCC 19606 is written as ATTC

19606 in line 346; nine types of antimicrobial agents are used in the work, but is written as nine antimicrobial agents in line 363.

Line 367-377, the knockout protocol is not written in the actual way of procedure.

At the first appearance of an abbreviation, the full name should be provided, such as OD. Besides, the description of OD₆₀₀ is not equal in the manuscript that it is also written as OD_{600nm} and OD600 in some sentences.

Line 402, for the unit of NPN, should it be 10 μm or 10 μM?

μl and μL are mixed used.

When writing the P value, P should be in italic. Moreover, sometimes it is shown in uppercase, but sometimes in lowercase.

Line 487, “μg ml⁻¹” is a wrong way of showing the unit.

Comments of Reviewer 1

General comments: This is a well-written paper. The hypothesis building is very solid and make sense. The data is very well-rounded and favorably tested the hypothesis. The concept and findings are of interest, but many opportunities exist to improve the clarity of the presentation. Especially compared to the considerable amount of data, the figure legend and the result part contain not enough information to support readers to go through those data and follow the hypothesis building.

Overall response to Reviewer 1: Thank you very much for your kind comments on our manuscript. We have carefully reviewed and re-annotated the figures and re-written/expanded figure legends, results, and discussion. In the revised version, the new text is highlighted in yellow, and the underline indicates the revised text is to identify better the changes made to the previous version. Our point-to-point response is provided below in red text. We hope these changes improve the clarity and accuracy of the presentation.

In what follows, we would like to answer the questions you mentioned and give a detailed account of the changes made to the original manuscript.

Comment 1: The importance section is a little bit redundant, it can be cut shorter or deleted since the content is somehow covered in abstract and discussion.

Response: Thanks for your valuable counsel. We have tried to shorten the IMPORTANCE section (Page 2, line 34-41).

Comment 2: Line 63-89, the introduction of CRISPR-Cas is not in a very good order and may lead to confusion. Try starting with overall introduction of CRISP-Cas in bacteria immune system, then lead to I-Fb CRISPR-CAS and I-Fb CRISPR-Cas3

Response: Thank you for your important reminder. We have revised the text to address your concerns and hope that it is now clearer. Logical flow:

Horizontal gene transfer (HGT) of multidrug resistance genes is an effective way to obtain multidrug resistance phenotype in *A. baumannii*. CRISPR-Cas system, as a bacterial innate immune system, effectively prevents HGT. The relationship between CRISPR-Cas system with physiology and antibiotic resistance. Quorum sensing (QS) leads to the relationship between CRISPR-Cas system and QS (Page 4, lines 43-92).

Comment 3: Line 91, the majority of strains do not have the presence of CRISP-Cas, so this title might be mislead.

Response: Thank you for pointing this out. We have changed "Presence of CRISPR-Cas systems in *A. baumannii* isolates." to "**Detection of CRISPR-Cas systems in *A. baumannii* isolates.**" (Page 6, line 94).

Comment 4: Line 95, define MDR (resistant to three or more classes of antibiotic) and sensitive.

Response: We are grateful for the suggestion, and this sentence was rephrased according to the comment, "**Among the 245 *A. baumannii* isolates tested, 16/245 (6.53%) and 20/245 (8.16%) isolates were resistant to only one or two of the nine classes antibiotics tested, respectively. Specifically, 209/245 (85.31%) were classified as multidrug-resistant (resistance to three or more classes of antibiotics, MDR).**" (Page 6, line 96-99).

Comment 5: line 112, if you want to show no differences between group, an endpoint statistical test is needed.

Response: We gratefully appreciate your valuable suggestion. A two-tailed paired t-

test is now added to the legend of Supplementary Figure 1. "A two-tailed paired t-test was used to analyze differences in endpoint OD₆₀₀ for growth curves."

(Supplementary Figure 1).

Comment 6: Line 114, the supplement table S3 which should support the conclusion is not well-organized and with no table title or description. You may reorganize the table S3 or only make conclusion draw from table 1.

Response: Thank you for your important reminder. We have added supplementary Tables S5-S11 of drug resistance for AB43 $\Delta cas1$, AB43 $\Delta cas3$, AB43 $\Delta csy1$, AB43 $\Delta csy2$, AB43 $\Delta csy3$, AB43 $\Delta csy4$, and AB43 $\Delta crisper$ antimicrobial tests (Supplementary Table 5-11).

Comment 7: line 139, Can't draw conclusion about drug resistance here, only resistance gene expression level.

Response: Since we did not express it clearly, we are sorry for your misunderstanding. Among 245 *A. baumannii* clinical isolates, only two strains with complete type I-F CRISPR-Cas systems resisted one category of antibiotics tested. We deleted any component or abolished the CRISPR-Cas system, rendering AB43 significantly resistant to most tested drugs. So we concluded that an intact CRISPR-Cas system could repress antibiotic resistance in *A. baumannii*. To remove the possible misunderstanding and misconception, we have modified the corresponding parts in the "Complete CRISPR-Cas system represses antibiotic resistance in *A. baumannii*." to make the expression more accurate and clearer (Page 8, line 129-139).

Comment 8: line 160 figure 4B and 4C clarify which genes are related to efflux pump and which are related to TCS.

Response: We are grateful for the suggestion. To be more clear and following the reviewer's concerns, we have added a brief description as follows: "Similarly, qRT-PCR results showed that the mRNA expression of genes related to the efflux pump, such as ABC transporters (*macB*, *emrB*) (1), major facilitator superfamily (MFS) (*craA*, *rpoB*, *tetB*, *abaO*), resistance-nodulation-cell division (RND) superfamily (*adeB*, *adeG*), and the small multidrug resistance (SMR) protein family (*abeS*, *abeM*) in the knockout strain were elevated (Figure 3B, C) (2)." (Page 9, line 156-161).

And the clarify of TCS related genes were add a brief description as follows: "TCS (*pmrA*, *pmrB*, *adeR*, *adeS*, *bfmS*, *bfmR*, *baeS*) (3)" (Page 14, line 252).

We have also revised the Figure legends corresponding to these two parts (Figure 3, Figure 7, Figure 8).

Comment 9: line 235 From figure 7F and 7G, if the ANOVA test of the two figure are done separately, the significancy can not be combined to get 62.5%.

Response: We fully agree with the reviewer. We now redraw the figure in the revised manuscript and provide a separate one-way ANOVA test for deletion strains. "Compared with AB43, more than half (5/8, 62.5%) of the deletion strains: AB43 Δ *crispr-cas*, AB43 Δ *cas3*, AB43 Δ *csy1*, AB43 Δ *csy3*, AB43 Δ *csy4*, showed significantly elevated of *abaI* mRNA (Figure 6C)" (Page 12, line 220-222).

Comment 10: line 240-244, the explanation for figure 8 is too vague, hard to understand what those gene in 8E-8L are. This paragraph is too general.

Response: We have taken care to address this crucial point raised by the reviewer and have tried to describe the clarify these drug-resistant genes precisely. "Similarly, RT-qPCR results showed that the most of drug resistance genes: ABC transporters (*macB*,

emrB (1), MFS (*craA*, *rpoB*, *tetB*, *abaQ*), RND superfamily (*adeJ*, *adeB*, *adeG*), the SMR protein family (*abeS*, *abeM*) (2), TCS (*pmrA*, *pmrB*, *adeR*, *adeS*, *bfmS*, *bfmR*, *baeS*) (3), biofilm formation (*ompA*, *ompW*, *lpsB*, *abaI*) (4), clinically significant cephalosporin resistance gene (*ampC*, *bla_{OXA-51-like}*) (5), and other genes (6-9) in the AB43Δ*crispr-cas* and AB43Δ*cas3* were raised (Figure 8C-J)." (Page 14, line 250-255).

"Moreover, RT-qPCR results showed that the mRNA expression of genes related to drug resistance, such as MFS (*craA*, *rpoB*), RND superfamily (*adeJ*), TCS (*pmrA*, *pmrB*, *adeS*, *bfmS*, *bfmR*) (3), biofilm formation (*ompA*, *lpsB*) (4), clinically significant cephalosporin resistance gene (*bla_{OXA-51-like}*) (5), and others (6-9) in the AB43Δ*abaI* were reduced (Figure 7)." (Page 12, line 226-230).

Comment 11: line 258 There is no figure 9.

Response: We are sorry for missing Figure 9. We added Figure 9 in the revised manuscript.

Comment 12: line396 Is there a washing step for permeability assessment?

Response: Yes. "After washing three times with PBS" (Page 23, line 444-445).

Comment 13: Line 765 I represents?

Response: Thank you for pointing this out. " I represents intermediate" (Table 2, Supplementary Table 5-11).

Comment 14: Line 838 Why separate 7F and 7G? Was the ANOVA test done separately in 7F and 7G?

Response: We are very sorry for our negligence in the original version. We now redraw the figure in the revised manuscript and provide a separate one-way ANOVA

test for deletion strains. "Compared with AB43, more than half (5/8, 62.5%) of the deletion strains: AB43 Δ *crispr-cas*, AB43 Δ *cas3*, AB43 Δ *csy1*, AB43 Δ *csy3*, AB43 Δ *csy4*, showed significantly elevated of *abaI* mRNA (Figure 6C)" (Page 12, line 220-222).

Comments of Reviewer 2

General comments: *In this manuscript, authors verified the function of CRISPR-Cas system in rendering A. baumannii antibiotic sensitivity through various ways, such as inhibiting hydrolyzing enzymes, efflux pumps and biofilm formation, production of ROS and H₂O₂, decreasing the activity of SOD and NADP. However, the reason of the above changes is concluded to the regulation axis of Cas3 - AbaI - efflux pumps. It seems not very persuasive that there is not enough evidence to prove that Cas3 is the core of CRISPR-Cas system to lead to the outcomes, since the deletion of other components in this system could also affect the expression of Cas3 and bring similar results. Moreover, changes of AbaI and efflux pumps are not the only reason for antibiotic sensitivity of A. baumannii. So, the function of CRISPR-Cas is not clearly explained by the current results.*

The manuscript is difficult to follow, many of the sentences need revision, and English needs an extensive proofreading. The results section includes a discussion of results, while the discussion section reports a repetition of results. In addition, the discussion section does not fully explain all the results obtained in this work. Both sections need to be revised. There is a common problem in the manuscript including figures and tables, that gene names should be all lowercase in italic, and protein names should be capitalized for the first letter and non-italic. Take AbaI as an example, its gene name is abaI, and protein name is AbaI. Misused of the names are all through the manuscript.

Overall response to Reviewer 2: Thank you for spending time reviewing our manuscript and providing us with a list of constructive comments. We understand that you have some significant concerns regarding CRISPR-Cas in *Acinetobacter*

baumannii contributing to antibiotic susceptibility. To address these concerns, some changes and experiments have been made.

There is some evidence to determine that Cas3 is the core of CRISPR-Cas system in regulating *abaI* mRNA degradation. First, among the single *cas* gene knockdown strains, the expression of *abaI* in AB43 Δ *cas3* was uppermost. Second, *cas3*-negative clinical strains had the highest resistance rates than *cas3*-positive in I-Fb strains. Third, the AB43 Δ *cas3* expression of the intrinsic *ampC* and *bla*_{OXA-51}-like was highest in all deletion strains. Additionally, qRT-PCR results showed that the most of drug resistance genes were raised in AB43 Δ *cas3*. By analogy with that result, Cui et al.'s results suggest that Cas3 targets and downregulates the expression of the operon related to QS, resulting in the expression of *Salmonella* biofilm forming-related genes being changed (10). The Cas3 of *P. aeruginosa* recognizes and cleaves the mRNA for the bacterial QS regulator LasR to dampen the recognition of toll-like receptor 4 (TLR4) (11). Another interesting phenomenon was that when one specific component of the CRISPR-Cas system was knocked out, the expression of *cas3* was suppressed. This further confirms our speculation that loss of either component will result in a decreased expression of *cas3* and eventually lead to drug resistance. A similar phenomenon was found in 4893 strains of *A. baumannii* collected in the NCBI (12).

(Page 16, line 293-305)

Several different pathways in *A. baumannii* may be employed to mediate the antibiotic resistance regulation by CRISPR-Cas. But, the pathway of CRISPR-Cas target endogenous *abaI* mRNA plays an important role. To begin with, CRISPR-Cas targets *abaI* to increase ROS. An activated TCA cycle will further accelerate the generation of ROS. In addition, CRISPR-Cas by targeting *abaI* to suppresses the efflux pump while suppressing the energy required to drive it. Furthermore, CRISPR-

Cas target *abaI* can also repress lipid A and biofilm formation. Finally, by targeting *abaI*, CRISPR-Cas also controls the most multiple drug resistance genes. (Figure 9).

We feel sorry for the inconvenience brought to the reviewer. We have carefully reviewed, proofread, and re-annotated the figures and re-written/expanded figure legends, results, and discussion. In the revised version, the new text is highlighted in yellow, and the underline indicates the revised text is to identify better the changes made to the previous version. Our point-to-point response is provided below in red text. We hope these changes improve the clarity and accuracy of the presentation.

In what follows, we would like to answer the questions you mentioned and give a detailed account of the changes made to the original manuscript.

Comment 1: *In some sentences, the expression is not rigorous. For example, in line 17-20, the factors of altered targets, decreased permeability, overexpression of efflux pumps, metabolic changes and biofilm formation could cause resistance phenotype in A. baumannii, but not the way of obtaining multidrug resistance genes. It's better to say "obtains multidrug resistance phenotype".*

Response: Thank you for pointing this out. We have changed "The ways this organism obtains multidrug resistance genes" to "The ways this organism obtains multidrug resistance phenotype" (Page 2, line 17)

Comment 2: *In line 26 and 65, authors say AB43 has a complete I-Fb CRISPR-Cas system. Please clarify what is the type of I-Fb, since according to the references, the type I CRISPR-Cas system are classified as Type I-F1, Type I-F2 (Ref. 25). Please use the official classification name and provide correct references.*

Response: Thank you for your reminding. According to the reference, the type I-F CRISPR-Cas system is classified as Type I-F1, Type I-F2, also known as Type I-Fa,

Type I-Fb (13). (Page 4, line 58)

Comment 3: *In line 79-80, the authors describe that QS activates type I-F CRISPR-Cas expression and CRISPR adaptation in P. aeruginosa. And in line 84-85, authors say that the purpose of this work is to investigate the role of the Type I-Fb CRISPR-Cas system in modulating QS operation in A. baumannii. What's the relationship between QS and Type I-Fb CRISPR-Cas system? Who regulates who? Or are they in a regulation circle? There should be some background about this knowledge.*

Response: Thanks for your valuable counsel. Considering the reviewer's suggestion, we have added the relationship between QS and CRISPR-Cas system. "On the other hand, when bacteria invade mammalian host cells, the Cas3 of P. aeruginosa recognizes and cleaves the QS regulator lasR mRNA to enhance its virulence (11)." (Page 5, line 83-85)

Comment 4: *Between line 91-101, please list the analysis of antibiotic susceptibility results of all the 245 clinical strains. The number/percentage of MDR and sensitive strains in 132 CRISPR-Cas negative isolates should also be shown in the results.*

Response: We gratefully appreciate for your valuable suggestion. Considering the reviewer's suggestion, we have added antibiotic susceptibility results in all the 245 clinical strains. "Of the 245 randomly collected A. baumannii clinical isolates, no isolate was susceptible to all 24 antibiotics (Figure 1A). Among the 245 A. baumannii isolates tested, 16/245 (6.53%) and 20/245 (8.16%) isolates were resistant to only one or two of the nine classes antibiotics tested, respectively. Specifically, 209/245 (85.31%) were classified as multidrug-resistant (resistance to three or more classes of antibiotics, MDR)." (Page 6, line 94-99)

The number/percentage of MDR and sensitive strains in 132 CRISPR-Cas negative isolates were added, too. "Additional, 132 isolates were not positive for a CRISPR-

Cas system, 4/132 (3.03%) and 8/132 (6.06%) isolates were resistant to only one or two of the nine classes antibiotics tested, respectively, and 120/132 (90.91%) were classified as MDR (Figure 1C)." (Page 7, line 111-114)

Comment 5: Line 98-101, should it be related between MDR and missing of component of the CRISPR-Cas system? Since most of the strains with incomplete or without CRISPR-Cas system are MDR.

Response: We fully agree with the reviewer. "To explore whether drug resistance in *A. baumannii* possessing an incomplete CRISPR-Cas system is associated with a specific Cas protein, we statistically analyzed the relationship between drug resistance phenotypes and cas genes in these 113 CRISPR-Cas-positive strains, and the results are shown in Table 1. We found that all cas genes-negative strains had significantly higher resistance rates than positive strains. I-Fa *csy3*-negative or I-Fb *cas3*-negative had the highest resistance rates in I-Fa and I-Fb cas genes-negative strains, respectively. In this regard, it is speculated that the incomplete CRISPR-Cas system, especially the loss of I-Fa *csy3* and I-Fb *cas3*, may affect antibiotic resistance in *A. baumannii*." (Page 7, line 116-124)

Comment 6: In Figure 1B, the subscripts of X axis are wrong, they should be I-Fa and I-Fb. Please also check whether they should be I-F1 and I-F2.

Response: Thank you so much for your careful check. We have made corrections. According to the reference, the type I-F CRISPR-Cas system is classified as Type I-F1, Type I-F2, also known as Type I-Fa, Type I-Fb (13). (Figure 1D)

Comment 7: In line 117-118, authors say that WT and all the gene rescue mutants were susceptible to all the 24 antibiotics. However, in Table 1, AB43 and the complement strain are resistant to piperacillin, and intermediate to cefotaxime and ceftriaxone. Please correct the results.

Response: Thank you so much for your careful check. We have changed "While the WT and all the gene rescue mutants were susceptible to all of these antibiotics" to "While the WT and all the gene rescue mutants were susceptible to most antibiotics," (Page 8, line 137)

Comment 8: Line 124, *bla*_{OXA-51-like} should be *bla*_{OXA-51-like}. Moreover, as shown in the results that the expression of *ampC* and *bla*_{OXA-51-like} were increased significantly in the Δ CRISPR-Cas mutant, why don't authors consider the inhibition function of CRISPR-Cas on these two enzymes, but give the conclusion only on regulation of efflux pumps?

Response: We are grateful for the suggestion. We make corrections to *bla*_{OXA-51-like}. Furthermore, in the Discussion section, we added the discussion about the inhibition function of CRISPR-Cas on *ampC* and *bla*_{OXA-51-like}. "CRISPR-Cas also represses drug-resistant related genes by targeting *abaI*. β -lactamase is an effective resistance mechanism of *A. baumannii* that can inactivate β -lactam antibiotics (14). Based on the sequence homology, β -lactamases were classified into four types: class A extended-spectrum β -lactamases (ESBL), class B Metallo- β -lactamases (MBL), class C β -lactamases (*AmpC*) and Class D β -lactamases (OXA) (15). All four types of β -lactamases were reported in *A. baumannii* (16). Compared to AB43, various resistance genes were expressed at elevated levels in AB43 Δ *crispr-cas*. It is striking that the intrinsic drug resistance gene *ampC*, often found in *A. baumannii* from China (17, 18), can be up to 200-fold. Another intrinsic drug resistance gene, *bla*_{OXA-51-like} was found up to 80-fold. Nevertheless, the expression of *ampC* did not change and *bla*_{OXA-51-like} significant reduction in AB43 Δ *abaI* and AB43 Δ *crispr-cas-abaI*. A similar phenomenon has been previously reported. According to Dou et al., AHLs generated by *A. baumannii* might increase the expression of drug-resistance genes

such as *bla*_{OXA-51-like}, *ampC*, *adeA*, and *adeB* (19)." (Page 19, line 351-363)

Comment 9: *Line 193-195, the explanation sounds conflict with the results. Since dysfunction of the CRISPR-Cas system could enhance biofilm formation, and at the same time dampen bacterial permeability, both two results will result in the increased resistance to antibiotics, thus show a synergistic effect with efflux pumps, but not limit the efflux of antibiotics.*

Response: Thank you for pointing this out. We have changed "Taken together, these results indicated that dysfunction of the CRISPR-Cas system could enhance AB43 biofilm biomass and dampen bacterial permeability, which in turn might limit the efflux of antibiotics." to "These results indicated that dysfunction of the CRISPR-Cas system could enhance AB43 biofilm biomass and dampen bacterial membrane permeability, which shows a synergistic effect with efflux pumps." (Page 11, line191-193)

Comment 10: *Line 240, authors say that several drug resistance factors are under that control of *AbaI* according to ref. 47. However, *AbaI* is not mentioned in this reference. BTW, authors did not say which factors are under the control of *AbaI*, thus the expression results could hardly be explained.*

Response: We are sorry for the inappropriate citation. We have corrected and cited the above studies. "Moreover, the QS system regulates bacterial luminescence, toxin production, disinfectants tolerance, motility, biofilm formation, spore formation, and drug resistance (20)." (Page 12, line 213-215)

Comment 11: *In Figure 8F, the *abaI* expression is increased in *AB43*Δ*CRISPR-Cas* mutant, that means *CRISPR-Cas* system represses the function of *AbaI*. And in Figure 8I, *adeB* expression is increased in *AbaI* knockout mutant, which shows another*

*negative regulation. In this case, downregulation of CRISPR-Cas could result in an upregulation of *abaI*, and finally lead to a decrease of *adeB* expression. However, in Figure 4C, the expression of *adeB* is increased for about 5 folds in Δ CRISPR-Cas mutant. How to explain the conflict results?*

Response: Thank you for your rigorous consideration. We also added explanations about this conflict results in the Discussion section. "The qRT-PCR results demonstrated that in AB43 Δ *abaI* and AB43 Δ *crispr-cas-abaI*, *adeR*, *baeS*, and some efflux pump-related genes such as *adeB* are increased. The results were not as expected. Additionally, the expression of *adeB* is increased by about five folds in AB43 Δ *crispr-cas* mutant. The results of these two studies seem to be contradictory. This is probably that although AB43 Δ *abaI* and AB43 Δ *crispr-cas-abaI* were designed to be AHL-deficient, subsequent experiments with these mutants may have been influenced by the presence of *abaI* homologs; *AbaI* is similar to the LuxI family of autoinducer synthases (21). In addition, it is indicated that a combination of multiple genes may produce efflux pump phenotypes." (Page 17, line 319-327)

Comment 12: *Moreover, if the results in Figure 8 are correct, expression of CRISPR-Cas will inhibit the activity of *AbaI*, which will boost the expression of *AdeB*, but not lead to ABC transporter inhibition as mentioned by authors in line 258.*

Response: Thank you for your significant reminding. We have changed "leading to ABC transporter inhibition because of the low level of QS (Figure 9). " to "Furthermore, we demonstrate that the I-Fb CRISPR-Cas system may target and degrade the *abaI* (QS synthase) mRNA, leading to drug resistance-related biological traits and genes being inhibited because of the low level of AHLs (Figure 9). " (Page 15, line 271-273)

Comment 13: In addition, in line 257-258, authors describe the relationship among *Cas3*, *AbaI* and ABC transporter, but all the qPCR results are obtained in Δ CRISPR-*Cas* mutant, and there is no evidence about the changes of *abaI* and *adeB* expression in *Cas3* knockout mutant.

Response: Considering the Reviewer's suggestion, we have added the results of the *abaI* and *adeB* expression changes in *cas3* knockout mutant. "Similarly, qRT-PCR results showed that the most of drug resistance genes: ABC transporters (*macB*, *emrB*) (1), MFS (*craA*, *rpoB*, *tetB*, *abaQ*), RND superfamily (*adeJ*, *adeB*, *adeG*), the SMR protein family (*abeS*, *abeM*) (2), TCS (*pmrA*, *pmrB*, *adeR*, *adeS*, *bfmS*, *bfmR*, *baeS*) (3), biofilm formation (*ompA*, *ompW*, *lpsB*, *abaI*) (4), clinically significant cephalosporin resistance gene (*ampC*, *bla*_{OXA-51-like}) (5), and other genes (6-9) in the AB43 Δ *crispr-cas* and AB43 Δ *cas3* were raised (Figure 8C-J)." (Page 14, line 250-255)

Comment 14: There are many mistakes in the manuscript, for example, ATCC 19606 is written as ATTC 19606 in line 346; nine types of antimicrobial agents are used in the work, but is written as nine antimicrobial agents in line 363.

Response: Thank you for pointing this out. We have made corrections. (Page 20, line 389; Page 21, line 406)

Comment 15: Line 367-377, the knockout protocol is not written in the actual way of procedure.

Response: We have re-written this part according to the reviewer's suggestion. "Briefly, using AB43 genomic DNA and PKD4 as templates, the upstream and downstream homology arms of the target fragment and the kanamycin cassette fragment with FRT site were amplified, respectively. Three PCR amplicons containing overlapping regions were assembled using overlap extension PCR with

specific primers (Table S4), and the resulting fragment was electroporated into competent AB43 carrying pAT04, which expresses the RecAB recombinase. Transformants were selected on LB plates containing 7.5 µg/ml kanamycin, and PCR confirmed integration of the resistance marker. To remove the kanamycin resistance cassette, electrocompetent mutants were transformed with pAT03 plasmid, which expresses the FLP recombinase. A loss of kanamycin resistance was observed in these colonies, confirmed by PCR, and sequenced using identification primers (Table S4)."

(Page 22, line 413-423)

Comment 16: At the first appearance of an abbreviation, the full name should be provided, such as OD. Besides, the description of OD600 is not equal in the manuscript that it is also written as OD600nm and OD600 in some sentences.

Response: Thank you for pointing this out. We have made corrections. "OD₆₀₀ (optical density at 600 nm)" (Page 22, line 429)

Comment 17: Line 402, for the unit of NPN, should it be 10 µm or 10 µM? µl and µL are mixed used. When writing the P value, P should be in italic. Moreover, sometimes it is shown in uppercase, but sometimes in lowercase.

Response: Thank you for pointing this out. We have made corrections and unified the concentration unit writing of NPN as 10 µM. (Page 23, line 449)

We have made unified the µL and *P* value.

Comment 18: Line 487, "µg ml⁻¹" is a wrong way of showing the unit.

Response: Thank you for pointing this out. We have made corrections. (Page 28, line 535)

Additional clarifications

In addition to the above comments, all spelling and grammatical errors pointed out by the reviewers have been corrected.

Other changes:

1. We changed the title "CRISPR-Cas3 in *Acinetobacter baumannii* Contributes to Antibiotic Susceptibility by Repressing Efflux Pumps in a Quorum-Sensing Dependent Manner" to "CRISPR-Cas in *Acinetobacter baumannii* Contributes to Antibiotic Susceptibility by Targeting Endogenous *AbaI*."
2. To make the logic more coherent and clearer, we have adjusted the order of some pictures.
3. We correct "the QS synthase gene *abaI* contains four regions matching the CRISPR array." to "QS synthase gene *abaI* contains one region matching the CRISPR array. The region from nucleotides (nt) 29 to 39 in *abaI* is partly matched with spacer 101 and repeats (Figure 6A)." (Page 12, line 215-217)

We tried our best to improve the manuscript and made some changes in the manuscript. These changes will not influence the content and framework of the paper. And here, we did not list the changes but marked them in red in the revised paper.

We appreciate for Editors/Reviewers' warm work earnestly and hope that the correction will meet with approval.

We look forward to hearing from you regarding our submission and responding to any further questions and comments you may have.

Once again, thank you very much for your comments and suggestions.

Sincerely,

Corresponding author:

Name: Guocai Li

E-mail: gcli@yzu.edu.cn

References

1. Greene NP, Kaplan E, Crow A, Koronakis V. 2018. Antibiotic Resistance Mediated by the MacB ABC Transporter Family: A Structural and Functional Perspective. *Front Microbiol* 9:950.
2. Abdi SN, Ghotaslou R, Ganbarov K, Mobed A, Tanomand A, Yousefi M, Asgharzadeh M, Kafil HS. 2020. *Acinetobacter baumannii* Efflux Pumps and Antibiotic Resistance. *Infect Drug Resist* 13:423-434.
3. Tierney AR, Rather PN. 2019. Roles of two-component regulatory systems in antibiotic resistance. *Future Microbiol* 14:533-552.
4. Colquhoun JM, Rather PN. 2020. Insights Into Mechanisms of Biofilm Formation in *Acinetobacter baumannii* and Implications for Uropathogenesis. *Front Cell Infect Microbiol* 10:253.
5. Eichenberger EM, Thaden JT. 2019. Epidemiology and Mechanisms of Resistance of Extensively Drug Resistant Gram-Negative Bacteria. *Antibiotics (Basel)* 6:37.

6. Ferrand A, Vergalli J, Pagès JM, Davin-Regli A. 2020. An Intertwined Network of Regulation Controls Membrane Permeability Including Drug Influx and Efflux in Enterobacteriaceae. *Microorganisms* 1:833.
7. Qin H, Lo NW, Loo JF, Lin X, Yim AK, Tsui SK, Lau TC, Ip M, Chan TF. 2018. Comparative transcriptomics of multidrug-resistant *Acinetobacter baumannii* in response to antibiotic treatments. *Sci Rep* 8:3515.
8. Selvaraj A, Valliammai A, Sivasankar C, Suba M, Sakthivel G, Pandian SK. 2020. Antibiofilm and antivirulence efficacy of myrtenol enhances the antibiotic susceptibility of *Acinetobacter baumannii*. *Sci Rep* 10:21975.
9. Trastoy R, Manso T, Fernández-García L, Blasco L, Ambroa A, Pérez Del Molino ML, Bou G, García-Contreras R, Wood TK, Tomás M. 2018. Mechanisms of Bacterial Tolerance and Persistence in the Gastrointestinal and Respiratory Environments. *Clin Microbiol Rev* 1:e00023-18.
10. Cui L, Wang X, Huang D, Zhao Y, Feng J, Lu Q, Pu Q, Wang Y, Cheng G, Wu M, Dai M. 2020. CRISPR-cas3 of *Salmonella* Upregulates Bacterial Biofilm Formation and Virulence to Host Cells by Targeting Quorum-Sensing Systems. *Pathogens* 9:53.
11. Li R, Fang L, Tan S, Yu M, Li X, He S, Wei Y, Li G, Jiang J, Wu M. 2016. Type I CRISPR-Cas targets endogenous genes and regulates virulence to evade mammalian host immunity. *Cell Res* 26:1273-1287.
12. Mortensen K, Lam TJ, Ye Y. 2021. Comparison of CRISPR-Cas Immune Systems in Healthcare-Related Pathogens. *Front Microbiol* 12:758782.
13. Hauck Y, Soler C, Jault P, Mérens A, Gérome P, Nab CM, Trueba F, Barges L, Thien HV, Vergnaud G, Pourcel C. 2012. Diversity of *Acinetobacter baumannii* in four French military hospitals, as assessed by multiple locus

variable number of tandem repeats analysis. PLoS One 7:e44597.

14. Peleg AY, Seifert H, Paterson DL. 2008. *Acinetobacter baumannii*: emergence of a successful pathogen. Clin Microbiol Rev 21:538-582.
15. Tooke CL, Hinchliffe P, Bragginton EC, Colenso CK, Hirvonen VHA, Takebayashi Y, Spencer J. 2019. β -Lactamases and β -Lactamase Inhibitors in the 21st Century. J Mol Biol 431:3472-3500.
16. Chang-Ro L, Hun LJ, Moonhee P, Seung PK, Kwon BI, Bae KY, Chang-Jun C, Chul JB, Hee LS. 2017. Biology of *Acinetobacter baumannii*: Pathogenesis, Antibiotic Resistance Mechanisms, and Prospective Treatment Options. Frontiers in Cellular & Infection Microbiology 13:55.
17. Wang H, Liu YM, Chen MJ, Sun HL, Xie XL, Xu YC. 2003. [Mechanism of carbapenems resistance in *Acinetobacter baumannii*]. Zhongguo Yi Xue Ke Xue Yuan Xue Bao 25:567-72.
18. Wei LH, Zhang J, Deng JJ, Zou FM, Liu G, Si XQ. 2004. [The isolation of *acinetobacter* strain from burn wound and the analysis of its antibiotic resistance]. Zhonghua Shao Shang Za Zhi 20:17-9.
19. Dou Y, Song F, Guo F, Zhou Z, Zhu C, Xiang J, Huan J. 2017. *Acinetobacter baumannii* quorum-sensing signalling molecule induces the expression of drug-resistance genes. Mol Med Rep 15:4061-4068.
20. Zhao X, Yu Z, Ding T. 2020. Quorum-Sensing Regulation of Antimicrobial Resistance in Bacteria. Microorganisms 17:425.
21. Jun, Zhu, Yunrong, Chai, Zengtao, Zhong, L., Shunpeng, Li, Winans. 2003. Agrobacterium Bioassay Strain for Ultrasensitive Detection of N-Acylhomoserine Lactone-Type Quorum-Sensing Molecules: Detection of Autoinducers in *Mesorhizobium huakuii*. Applied & Environmental

Microbiology 69:6949-53.

July 14, 2022

Prof. Guocai Li
Yangzhou University
Pathogen Biology and Immunology
11 Huai-hai Road
Yangzhou, Jiangsu 225001
China

Re: Spectrum00829-22R1 (CRISPR-Cas in *Acinetobacter baumannii* Contributes to Antibiotic Susceptibility by Targeting Endogenous *AbaI*)

Dear Prof. Guocai Li:

Your manuscript has been accepted, and I am forwarding it to the ASM Journals Department for publication. You will be notified when your proofs are ready to be viewed.

Sincerely,

Monica Garcia-Solache
Editor, Microbiology Spectrum
